# Concurrent Reinforcement Learning with Aggregated States via Randomized Least Squares Value Iteration

**Yan Chen** [1]  **Qinxun Bai** [2]  **Yiteng Zhang** [3]  **Maria Dimakopoulou** [4]  **Shi Dong** [5]  **Qi Sun** [6]  **Zhengyuan Zhou** [6 7]

## Abstract

Designing learning agents that explore efficiently in a complex environment has been widely recognized as a fundamental challenge in reinforcement learning. While a number of works have demonstrated the effectiveness of techniques based on randomized value functions on a single agent, it remains unclear, from a theoretical point of view, whether injecting randomization can help a society of agents *concurently* explore an environment. The theoretical results established in this work tender an affirmative answer to this question. We adapt the concurrent learning framework to *randomized least-squares value iteration* (RLSVI) with *aggregated state representation*. We demonstrate polynomial worst-case regret bounds in both finite- and infinite-horizon environments. In both setups the per-agent regret decreases at an optimal rate of $\Theta\left(\frac{1}{\sqrt{N}}\right)$, highlighting the advantage of concurrent learning. Our algorithm exhibits significantly lower space complexity compared to (Russo, 2019) and (Agrawal et al., 2021). We reduce the space complexity by a factor of $K$ while incurring only a $\sqrt{K}$ increase in the worst-case regret bound, compared to (Agrawal et al., 2021; Russo, 2019). Interestingly, our algorithm improves the worst-case regret bound of (Russo, 2019) by a factor of $H^{1/2}$, matching the improvement in (Agrawal et al., 2021). However, this result is achieved through a fundamentally different algorithmic enhancement and proof technique. Additionally, we conduct numerical experiments to demonstrate our theoretical findings.

## 1. Introduction

The field of reinforcement learning (RL) is dedicated to designing agents that interact with an unknown environment, aiming to maximize the total amount of reward accumulated throughout the interactions (Sutton & Barto, 2018). When the environment is complex yet the learning budget is limited, an agent has to efficiently explore the environment, giving rise to the well-known exploration-exploitation trade-off. A large body of works in the RL literature have addressed the challenge related to smartly balancing this trade-off. Among the many proposed methods, algorithms based on randomization are receiving growing attention, both theoretically (Russo & Van Roy, 2014; Fellows et al., 2021) and empirically (Osband et al., 2016; Janz et al., 2019; Dwaracherla et al., 2020), due to their effectiveness and potential scalability in large applications.

Randomized least-squares value iteration (RLSVI) represents one example of such randomization-based algorithms (Osband et al., 2019). On a high level, RLSVI injects Gaussian noise into the rewards in the agent's previous trajectories, and allows the agent to learn a randomized value function from the perturbed dataset. With judicious noise injection, the resultant value function approximates the agent's posterior belief of state values. By acting greedily with respect to such randomized value function, the agent effectively executes an approximated version of posterior sampling for reinforcement learning (PSRL), whose efficacy has been substantiated in previous works (Osband et al., 2013; Russo & Van Roy, 2014; Xu et al., 2022). Compared with PSRL, RLSVI circumvents the need of maintaining a model of the environment, severing huge computational cost. Since its advent, RLSVI has been studied extensively in theoretical contexts, such as (Russo, 2019) and (Ishfaq et al., 2021).

In this work, we look into RLSVI from the perspective of concurrent learning (Silver et al., 2013). Specifically, concurrent RL studies the problem where a cohort of agents interact with one common environment, yet are able to share their experience with each other in order to jointly improve their decisions. Such a collaborative setting has useful applications in a variety of realms, such as robotics (Gu et al., 2017), biology (Sinai et al., 2020), and recommendation

---

[1]Duke University [2]Horizon Robotics Inc [3]UC Berkeley [4]Uber [5]Google DeepMind [6]New York University [7]Arena Technologies. Correspondence to: Zhengyuan Zhou <z@arena-ai.com>.

*Proceedings of the $42^{nd}$ International Conference on Machine Learning*, Vancouver, Canada. PMLR 267, 2025. Copyright 2025 by the author(s).

systems (Agarwal et al., 2016). For instance, a team of drones may be cooperatively executing a search and rescue mission in a given terrain, and cooperatively learning to identify the locations of assets requires efficient concurrent RL. Another example is a team of virtual agents conducting circuit design, proceeding in parallel the design-circuit evaluation (via a simulator)-redesign loop. Here, they are cooperatively exploring the vast and complex design space, aiming to quickly identify an optimal design that meets all the sepc requirements[1]. Importantly, although all agents share the identical goal, coordination between agents is nontrivial. In fact, as shown in (Dimakopoulou & Van Roy, 2018; Dimakopoulou et al., 2018), a poorly coordinated multi-agent algorithm can drastically undermine the overall learning performance. Results in the existing theoretical literature on concurrent RL have demonstrated that PSRL (Osband & Van Roy, 2017; Kim, 2017; Osband & Van Roy, 2014) in a coordinated style is provably efficient (Chen et al., 2022), compared to the earlier cooperative UCRL type of algorithms [2]. As pointed out in (Chen et al., 2022), "these sample-complexity guarantees notwithstanding, concurrent UCRL algorithms suffer from the critical disadvantage of no coordinated exploration: since the upper confidence bounds computed by aggregating all agents' data are the same the entire team, each agent would follow the exact same policy, thereby yielding no diversity in exploration." However, despite the exploration advantage offered by concurrent Thompson sampling given in a very recent work (Chen et al., 2022), it remains unclear whether the same level of efficiency can be extended to cooperative model-free agents, which are computationally more scalable.

**Our Contributions** We model the environment as a Markov decision process (MDP) with $\Gamma$ aggregated states, and show theoretically that a cohort of $N$ agents, concurrently running RLSVI and sharing interaction trajectories with each other, are able to efficiently improve their joint

---

[1]For instance, Arena, a world leading startup that specializes in applying AI to shorten the hardware design loop, utilizes RL to speed up the circuit design, test and debugging cycles.

[2]It is perhaps not surprising that UCRL algorithms are the first family of algorithms that have been adapted to the concurrent RL settings (Guo & Brunskill, 2015) and (Pazis & Parr, 2016). In particular, concurrent UCRL has been analyzed for the sample complexity performance measure (i.e. how many samples are needed to learn an $\epsilon$-optimal policy) under both finite action space setting and infinite action space setting. More specifically, (Guo & Brunskill, 2015) provided a high-probability bound of $\tilde{O}(\frac{S^2 A}{\epsilon^3} + \frac{SAn}{\epsilon})$ for the sample complexity with $n$ agents interacting in the environment. Their algorithm was extended from MBIE(see (Strehl & Littman, 2008)), with a single agent performing concurrent RL across a set of $n$ infinite-horizon MDPs. The results there show that with sharing samples from copies of the same MDP a linear speedup in the sample complexity of learning can be achieved. But no regret bound was derived there for concurrent RL.

performance towards the optimal policy in the environment. The efficiency is established through regret analysis. Additionally, for aggregated state representation setup, the performance typically will not converge to optimal solution if $\epsilon > 0$, and our result establishes that the cumulative performance loss is no greater than $O(\epsilon)$ per period per agent. This is consistent with the findings of (Dong et al., 2019b) for single-agent scenario. Our worst-case regret bound improves upon (Russo, 2019) by a factor of $H^{1/2}$, matching the improvement in (Agrawal et al., 2021) but achieved through a fundamentally different technique and proof approach.

One major difference between our algorithm and those from current literature of RLSVI in tabular setup is that, the algorithms in (Russo, 2019; Agrawal et al., 2021) require storing the historical trajectories from the very beginning while our algorithm only stores the trajectories from the last episode for the finite-horizon case or the last pseudo-episode (defined in Section 4) for the infinite-horizon case. This is because for concurrent setup, the computational cost of storing all historical data scales at least linear in $KHN$ for finite horizon and $TN$ for infintie horizon, which is infeasible for problems of practical scale.

In both cases, the regret dependence on the total number of samples is optimal, signaling well-coordinated information sharing among the agents. To our best knowledge, this work presents the first theoretical analysis of a model-free concurrent RL algorithm with aggregated states.

Besides theoretical contributions, our analysis also sheds light on the empirical role of discount factor in RL. In practical applications, a discount factor is usually not subsumed under the environment definition. Rather than prescribing a specific learning target that involves a decaying sequence of rewards, discount factors often simply function as a tuning parameter of the algorithm that keeps the value updates stable. In this work, similar as in (Xu et al., 2022), when the decision horizon is infinite, the discount factor only shows up in the algorithm and does not appear in the learning target. The sole purpose of introducing a discount factor is to decompose the single stream of interactions into "pseudo-episodes" that facilitate decision-making. Such view aligns with the authentic role of discount factor in agent design, and distinguishes this work from earlier RL theory literature on discounted regret such as (Dong et al., 2019a). While (Xu et al., 2022) focuses on PSRL, this paper is the first to extend the same view to a model-free algorithm that enjoys empirical successes.

**Related Work** *Multi-agent Reinforcement Learning* (MARL) has been widely studied to address problems in a variety of applications like robotics, telecommunications and e-commerce (Buşoniu et al., 2010). Scenarios of MARL

include fully cooperative (Abed-alguni et al., 2015; Zhang et al., 2010), fully competitive (Bansal et al., 2017; Wang & Klabjan, 2018) and other more general settings (Ryu et al., 2021; Lowe et al., 2017). Under these settings, a group of agents share and interact with a common random environment (Shoham & Leyton-Brown, 2008; Vlassis, 2022; Weiss, 1999). There are certain challenges of multi-agent systems as pointed out by (Zhang et al., 2021). For example, the agents may have non-unique learning goals (Shoham et al., 2003). Besides, the concurrent learning structure of MARL problem can cause the environment faced by each individual to be non-stationary. Particularly in some settings, the action taken by one agent affects the reward of other opponent agents and the evolution of the state (Zhang et al., 2021). Furthermore, to handle non-stationarity of the problems, the agents may need to form the joint action space, and the dimension the action spaces can increase exponentially with the number of agents (Kulkarni & Tai, 2010).

We study the MARL problem under a concurrent learning framework involving homogeneous agents with a common learning goal. The agents interact with the unknown environment independently in parallel, and draw actions from a commonly shared action space. They communicate and share information according to some strategically devised schedule. All the agents share the same reward functions according to the states and actions they take. This is different from the Markov game setup (Szepesvári & Littman, 1999; Littman, 2001) where the reward is influenced by the joint action of all the agents in the system.

We apply concurrent learning (Min et al., 2023; Dubey & Pentland, 2021) concept with the Randomized Least-Squares Value Iteration (RLSVI) learning framework with aggregated state representations (Dong et al., 2019b; Van Roy, 2006). On the one hand, RLSVI leverages random perturbations to approximate the posterior, applying frequentist regret analysis in the tabular MDP setting (Osband et al., 2016), inspiring works that focus on theoretical analyses to improve worst-case regret in tabular MDPs (Russo, 2019; Agrawal et al., 2021) and linear settings (Zanette et al., 2020; Ishfaq et al., 2023; Dann et al., 2021). On the other hand, while a large body of literature is established on tabular representations (Agarwal et al., 2020; Azar et al., 2011; Auer et al., 2008; Jiang et al., 2017; Jin et al., 2018; Osband et al., 2013; 2016; Strehl & Littman, 2008), aggregated state representation offers an approach to reduce statistical complexity given the fact that the data requirement and learning time scales with the number of state-action pairs in tabular representations. In this paper, we present a concurrent version of the randomized least-squared value iteration algorithm with aggregated states in finite-horizon and infinite-horizon settings, and we provide their corresponding worst-case regret bounds and numerical performances.

As an outline of the rest of the paper, in Section 2, we define the finite-horizon case and introduce a concurrent learning framework with aggregated states. In Section 3, we summarize the finite-horizon concurrent RLSVI algorithm in Algorithm 3 and provide its worst-case regret bound in Theorem 2. Section 4 focuses on the infinite-horizon concurrent learning framework with aggregated states, with the infinite-horizon concurrent RLSVI algorithm summarized in Algorithm 4. The corresponding worst-case regret bound is provided in Theorem 7 of Section 5. Numerical results of both algorithms are reported in Section 6. We also include Table 1 in Appendix A comparing our approach with prior work.

## 2. Finite-Horizon Concurrent Learning

In this section, we consider a finite-horizon *Markov Decision Process* (MDP) $M = (H, \mathcal{S}, \mathcal{A}, P, R)$. There are $N$ agents interacting with the same environment across $K$ episodes. Each episode contains $H$ periods. For episode $k \in [K]$, period $h \in [H]$, and agent $p \in [N]$, we use $s_{k,h}^p$ to denote the state that the agent resides in, $a_{k,h}^p$ the action that the agent takes, and $r_{k,h}^p = r(s_{k,h}^p, a_{k,h}^p)$ the reward that it receives, where $r : \mathcal{S} \times \mathcal{A} \mapsto [0, 1]$ is a deterministic reward function. Let the information set $\mathcal{H}_k = \{(s_{k,h}^p, a_{k,h}^p, r_{k,h}^p) : h \in [H]\}$ be the trajectory during episode $k$ for all the agents. The transition kernel $P$ is defined as $P_{h,s,a}(s') = \mathbb{P}\left(s_{h+1} = s' \big| a_h = a, s_h = s\right)$. The expected reward that an agent receives in state $s$ when it follows policy $\pi$ at step $h$ is represented by $R_{h,s,\pi(s)} = \mathbb{E}\left[\sum_{a \in \mathcal{A}} \pi_h(a|s) \cdot r(s, a)\right]$. We assume that all agents start from a deterministic initial state $\mathbf{s}_1 = \{s_1^p\}_{p \in [N]}$ with $s_{k,1}^p \equiv s_1^p, \forall k, p$. In this work we consider deterministic rewards, which can be viewed as mappings from $\mathcal{S}$ to $\mathcal{A}$, but all our results apply to the environments with bounded rewards without change. We say that agent $p \in [N]$ follows policy $\pi$ if for all $h \in [H]$, $a_h^p = \pi_h(s_h^p)$. We use $V_h^\pi \in \mathbb{R}^S$ to denote the value function associated with policy $\pi$ in period $h \in [H]$, such that

$$V_h^\pi(s) = \mathbb{E}\left[\sum_{j=h}^{H} R_{j,s_j,\pi_j(s_j)} \big| s_h = s\right],$$

where the expectation is taken over all possible transitions, and we set $V_{H+1}^\pi(s) = 0$ for all $s \in \mathcal{S}$. The optimal value function is denoted as $V_h^*(s) = \max_{\pi \in \Pi} V_h^\pi(s)$, which is the value function associated with the optimal policy. For all $s \in \mathcal{S}, h \in [H]$, and policy $\pi$, the value function is the unique solution to the Bellman equations

$$V_h^\pi(s) = R_{h,s,\pi(s)} + \sum_{s \in \mathcal{S}} P_{s,h,\pi(s)}(s') V_{h+1}^\pi(s').$$

When $\pi$ is the optimal policy $\pi^*$, there should be $V_h^{\pi^*}(s) = V_h^*(s)$, and we have

$$V_h^*(s) = R_{h,s,\pi^*(s)} + \sum_{s \in \mathcal{S}} P_{s,h,\pi^*(s)}(s') V_{h+1}^*(s').$$

For each policy $\pi$, we also define the *state-action value function* of *Q-function* of state-action pair $(s, a)$ as the expected return when agent takes action $a$ at state $s$, and then follows policy $\pi$, so that

$$Q_h^\pi(s, a) = R_{h,s,a} + \mathbb{E}\left[\sum_{j=h}^{H} R_{j,s_j,\pi_j(s_j)} \Big| s_h = s, a_h = a\right].$$

Correspondingly, we define

$$Q_h^*(s, a) = R_{h,s,a} + P_h V_{h+1}^*(s, a),$$

where we use the notation $P_h V(s, a) = \mathbb{E}_{s' \sim P_h^{s,a}}[V(s')]$. Thus by definition $Q_h^*(s, a)$ is the maximum realizable expected return when the agent starts from state $s$ and takes action $a$ at period $h$. From the optimality of $V^*$, we have

$$V_h^*(s) = \max_{a \in \mathcal{A}} Q_h^*(s, a), \ \forall h \in [H], s \in \mathcal{S}.$$

Furthermore, under the assumption that the reward is bounded between 0 and 1, we have

$$0 \le V_h^\pi \le V_h^* \le H, \ \forall h \in [H], \pi \in \Pi.$$

**Regret under Finite Horizon Case**   The goal of an RL algorithm is for the agents to learn a good policy through consecutively interacting with the random environment, without prior knowledge about the transition probability $P$ and the reward $R$. Formally, given $\pi = \{\pi^{kp}\}_{k \in [K], p \in [N]}$, with each agent $p \in [N]$ taking policy $\pi^{kp}$ during each episode $k \in [K]$, the cumulative expected regret incurred over $K$ periods and $N$ agents is defined as

$$\text{Regret}(M, K, H, N, \pi) = \sum_{p=1}^{N} \sum_{k=1}^{K} V_1^*(s_1^p) - V_{\pi^{kp}}(s_1^p).$$

$$(1)$$

**Empirical Estimation**   We examine two types of empirical estimation methods below. The first method stores data from a single episode only, meaning the empirical counts used by the agents to evaluate policies are derived from just one episode. We take this limitation into account because, as the number of agents $N$ increases, storing all historical data becomes practically infeasible. For the second method, we store all historical data; specifically, at each episode $k$, the buffer retains data from all episodes prior to $k$. This results in a space complexity of $O(KHN)$.

For the first empirical estimation method, define $n_{k,h}(s, a)$ to be the number of times action $a$ has been sampled in state $s$, period $h$ during episode $k$ by all the agents $p \in [N]$:

$$n_{k,h}(s, a) = \sum_{p=1}^{N} \mathbf{1}\left\{(s_{k,h}^p, a_{k,h}^p) = (s, a)\right\}.$$

Define the empirical mean reward for period $h$ during episode $k$ by

$$\hat{R}_{h,s,a}^k = \frac{\sum_{p=1}^{N} \mathbf{1}\left\{(s_{k-1,h}^p, a_{k-1,h}^p) = (s, a)\right\} r_{k-1,h}^p}{n_{k-1,h}(s, a)},$$

$$(2)$$

and $\forall s' \in \mathcal{S}$, define the empirical transition probabilities for period $h$ during episode $k$ as

$$\hat{P}_{h,s,a}^k(s')$$
$$= \frac{\sum_{p=1}^{N} \mathbf{1}\left\{(s_{k-1,h}^p, a_{k-1,h}^p, s_{k-1,h+1}^p) = (s, a, s')\right\}}{n_{k-1,h}(s, a)}.$$

$$(3)$$

If $(h, s, a)$ is never sampled during episode $k - 1$, we set $\hat{R}_{h,s,a}^k = 0 \in \mathbb{R}$ and $\hat{P}_{h,s,a}^k = 0 \in \mathbb{R}^S$. Note that $\hat{R}^k$ and $\hat{P}^k$ are computed from the trajectory from episode $k - 1$.

The second method stores all historical data up to the current episode, with the reward and transition probability estimations defined as:

$$\hat{R}_{h,s,a}^{k,\texttt{full}} = \frac{\sum_{i=0}^{k-1} \sum_{p=1}^{N} \mathbf{1}\left\{(s_{i,h}^p, a_{i,h}^p) = (s, a)\right\} r_{i,h}^p}{\sum_{i=0}^{k-1} n_{i,h}(s, a)},$$

$$\hat{P}_{h,s,a}^{k,\texttt{full}}(s')$$
$$= \frac{\sum_{i=0}^{k-1} \sum_{p=1}^{N} \mathbf{1}\left\{(s_{i,h}^p, a_{i,h}^p, s_{i,h+1}^p) = (s, a, s')\right\}}{\sum_{i=0}^{k-1} n_{i,h}(s, a)}.$$

### 2.1. Aggregated-state Representations

Many RL algorithms aim to estimate the value of each state-action pair (e.g. under a tabular representation), but this can be infeasible in some setup where $SA$ is large, since both the required sample size and computational cost will scale up at least linearly in $SA$. One alternative approach is to consider *aggregated-state representations* (Dong et al., 2019b; Wen & Van Roy, 2017; Jiang et al., 2017), which reduces complexity and can accelerate learning by focusing on aggregated state-action pairs. This method partitions the space of state-action pairs into $\Gamma$ blocks, each block can be viewed as an aggregate state, so that the value function representation only needs to maintain one value estimate per aggregated state. Formally, let $\Phi$ be the set of all aggregated states, and let $\phi_h : \mathcal{S} \times \mathcal{A} \to \Phi$ be the mapping from state-action pairs to aggregated states at period $h$. Without loss of generality, we let $\Phi = [\Gamma]$. We define the aggregated representation as follows:

**Definition 1.** *We say that $\{\phi_h\}_{h=1}^{H}$ is an $\epsilon$-error aggregated state-representation (or $\epsilon$-error aggregation) of an MDP, if for all $s, s' \in \mathcal{S}$, $a, a' \in \mathcal{A}$ and $h \in [H]$ such that $\phi_h(s, a) = \phi_h(s', a')$, we have*

$$|Q_h^*(s, a) - Q_h^*(s', a')| \leq \epsilon.$$

When $\epsilon = 0$ in Definition 1, we say that the aggregation is sufficient, and one can guarantee that an algorithm finds the optimal policy as $K \to \infty$. When $\epsilon > 0$, there exists an MDP such that no RL algorithm with aggregated state representation can find the optimal policy (Van Roy, 2006). In this case, the best we can do is to approximate the optimal policy with the suboptimality bounded by a function of $\epsilon$.

## 3. Finite-horizon Algorithm and Regret Bound

The concurrent RLSVI algorithms for the finite-horizon case are Algorithm 1 and Algorithm 3.

**Tradeoff: Regret Reduction vs. Sample Complexity**  Algorithm 1 and Algorithm 3 reflect a trade-off between reducing the worst-case regret and the consideration for the sample complexity in practice. Algorithm 1 stores all historical data, while Algorithm 3 retains only the data from the previous episode. Although Algorithm 1 achieves lower regret due to its larger data storage, it becomes impractical when $N$ is large. Therefore, for practical reasons, we use Algorithm 3 in our numerical experiments. We first describe Algorithm 3, with Algorithm 1 following a similar structure, differing only in the buffer size (i.e. the amount of data stored for policy evaluation).

We initialize the all the aggregated state values as $H$, i.e. $\hat{Q}_h^p(\gamma) = H$ for all $h \in [H]$. At the beginning of each episode, all the agents restart at initial states $\mathbf{s}_1 = \{s_1^p\}$, with agent $p$ starting from state $s_1^p$. During pre-round (episode 0), each agent randomly samples their initial trajectory $\{s_{0,1}^p, a_{0,1}^p, r_{0,1}^p, \ldots, s_{0,H}^p, a_{0,H}^p, r_{0,H}^p\}_{p=1}^{N}$, with $s_{0,1}^p = s_1^p$. During each episode $k \in [K]$, each agent samples a random vector with independent components $w^{kp} \in \mathbb{R}^{HSA}$, where $w^{kp}(h, s, a) \sim \mathcal{N}(0, \sigma_k^2(h, s, a))$ and $\sigma_k(h, s, a) = \sqrt{\frac{\beta_k}{N_{k-1,h}(\phi_h(s,a))+1}}$, where $\beta_k$ is a tuning parameter, $N_{k-1,h}(\phi_h(s,a))$ is the total number of times that aggregated state $\phi_h(s, a)$ is reached at period $h$ across all agents during episode $k-1$. Given $w^{kp}$, we construct a randomized perturbation of the empirical MDP for agent $p$ as

$$\overline{M}^{kp} = (H, \mathcal{S}, \mathcal{A}, \hat{P}^k, \hat{R}^k + w^{kp}), \quad (4)$$

where the empirical distributions $\hat{R}^k$, $\hat{P}^k$ are computed as in (2) and (3). During each episode $k \in [K]$, the data set $\tilde{D}_{kh}^p$ contains perturbation of samples from episode $k-1$ for time period $h$ used by agent $p$.

Each agent computes the values for aggregated states using a backward approximation, where during episode $k$, the uncapped value of aggregated state $\gamma$ during period $h$ computed by agent $p$ is

$$\begin{aligned}
&\bar{Q}_{k,h}^p(\gamma) \\
&= \underset{Q \in \mathbb{R}}{\arg\min} \, \mathcal{L}(Q | \hat{Q}_{k,h+1}^p, \tilde{D}_{kh}^p, \alpha_{N_{k-1,h}(\gamma)}, \xi_{N_{k-1,h}(\gamma)}) \\
&\qquad + \|Q - \alpha_{N_{k-1,h}(\gamma)} \tilde{Q}_{kh}^p\|_2^2,
\end{aligned}$$

where $\xi_{N_{k-1,h}(\gamma)}$ is defined as (7) for $n = N_{k-1,h}(\gamma)$, and we set terminal values as $\hat{Q}_{k,H+1}^p(\gamma) = 0$, and the regularization noise $\tilde{Q}_{kh}^p \in \mathbb{R}^{S \times A}$ is an independently sampled random vector, such that for each $(s, a) \in \mathcal{S} \times \mathcal{A}$, $\tilde{Q}_{kh}^p(s, a) \sim \mathcal{N}\left(0, \frac{\beta_k}{1+N_{k,h}(\phi_h(s,a))}\right)$, $\beta_k$ is a tuning parameter, and

$$\begin{aligned}
&\mathcal{L}(Q \mid Q_{\text{next}}, \mathcal{D}, \alpha, \xi) \\
&= \sum_{(s,a,r,s') \in \mathcal{D}} \{Q - \xi - (1 - \alpha)Q_{k-1,h}(\phi_h(s,a)) \\
&\qquad - \alpha(r + \max_{a' \in \mathcal{A}} Q_{\text{next}}(s', a'))\}^2.
\end{aligned} \quad (5)$$

Here the regularized loss function defined in (5) is such that $Q(\gamma)$ as the computed value of aggregated state $\gamma = \phi_h(s, a)$ for some pair $(s, a)$ approximates

$$(1 - \alpha_{N_{k-1,h}(\gamma)})Q_{k-1,h} + \alpha_{N_{k-1,h}(\gamma)}(r + \max_{a' \in \mathcal{A}} Q_{\text{next}}(s', a')).$$

And since $\alpha_{N_{k-1,h}(\gamma)} = \frac{1}{1+N_{k-1,h}(\gamma)}$ as defined in Theorem 2, we see that when $N_{k-1,h}(\gamma)$ increases, the algorithm puts more weight on the value learned from the previous episode. At the end of episode $k$, after each agent interacts with the environment, the algorithm takes a weighted average of the values learned by each agent $p \in [N]$, by taking

$$\hat{Q}_{k,h}(\gamma) = \frac{1}{N_{k,h}(\gamma)} \sum_{p=1}^{N} \mathbf{1}\{\phi_h(s_{k,h}^p, a_{k,h}^p) = \gamma\} \hat{Q}_{k,h}^p(\gamma) \quad (6)$$

for each $\gamma \in \Gamma$, where $N_{k,h}(\gamma)$ is the total number of times that aggregated state $\gamma$ appears during episode $k$ period $h$.

Given tuning parameter $\beta = \{\beta_k\}_{k \in \mathbb{N}}$, we denote Algorithm 3 by RLSVI$_\beta$. While Algorithm 3 is based on the RLSVI algorithm of (Russo, 2019), there are some notable differences. The algorithm of (Russo, 2019) is with single-agent, and at each time period $h$ during episode $k$, the agent needs to keep all the trajectory prior to episode $k$, which can be infeasible for multiple agents, because the space can grow very fast. In our algorithm, we only keep the historical data of the last episode, to make the algorithm computationally feasible. This leverages the fact that with $N$ agents interacting with the random environment, the information within one single episode is already rich. Algorithm 3 is also related to the aggregated-state algorithm

proposed by (Dong et al., 2019b), where the authors apply the aggregated-state idea to an optimisitic variant of Q-learning on a fixed-horizon episodic Markov decision process based on the previous UCB-type result by (Jin et al., 2020). Our work incorporates this idea in the concurrent randomized least square value iteration.

Algorithm 1 follows a similar structure, with $N_{k,h}(\phi_h(s,a))$ redefined as the total number of times the aggregated state $\phi_h(s,a)$ is visited at period $h$ across all agents up to and including episode $k$. We denote it by $\text{RLSVI}_\beta^{\text{full}}$ to indicate that Algorithm 1 uses the full history up to the current episode.

With the employment of aggregated state and the modified loss function for the learning process, the result highlights that the per-agent regret decreases at a rate of $\Theta\left(\frac{1}{\sqrt{N}}\right)$. The concurrent learning algorithm for the finite-horizon case is Algorithm 3. The worst-case regret bound for Algorithm 3 is provided in the next section.

### 3.1. Worst-case Regret Bound

Let $\mathcal{M}$ be the set of MDPs with episode number $K$, horizon $H$, state space size $S$, action space size $A$, transition probabilities $P$, and rewards $R$ bounded in $[0, 1]$. Let $N$ be the number of agents interacting in the same environment. We use $M = (K, H, \mathcal{S}, \mathcal{A}, P, R)$ to denote an MDP in $\mathcal{M}$.

Suppose $\{\phi_h\}_{h \in [H]}$ is an $\epsilon$-error aggregation (defined as in Definition 1) of the underlying MDP. For a tuning parameter sequences $\beta = \{\beta_n\}_{n \in \mathbb{N}}, \alpha = \{\alpha_n\}_{n \in \mathbb{N}}, \xi = \{\xi_n\}_{n \in \mathbb{N}}$ where $\beta_n = \frac{1}{2} H^3 \log(2H\Gamma n)$, $\alpha_n = \frac{1}{1+n}$, and

$$
\begin{aligned}
\xi_n \quad &= \epsilon + \frac{2\alpha_n H \sqrt{\log(2KHN/\delta)}}{\sqrt{\max\{n, 1\}}} \\
&+ \frac{2\alpha_n \sqrt{\beta_{k-1} \log(2KHN/\delta)}}{\sqrt{(n+1)\max\{n, 1\}}}
\end{aligned}
\tag{7}
$$

We now provide our main results for the finite-horizon case. As explained, Algorithm 1 stores all historical data, while Algorithm 3 retains only the previous episode, making it straightforward that Algorithm 1 has a space complexity of $O(KHN)$, while Algorithm 3 has a space complexity of $O(HN)$, and their worst-case regret bounds are

**Theorem 2.** *Algorithm 1 has a worst-case regret bound*

$$
\begin{aligned}
\sup_{M \in \mathcal{M}} \; &\text{Regret}(M, K, N, \pi, \text{RLSVI}_{\beta,\alpha,\xi}^{\text{full}}) \\
&\leq \widetilde{O}(\epsilon\sqrt{K}HN + H^{5/2}\Gamma\sqrt{KN}).
\end{aligned}
\tag{8}
$$

*Algorithm 3 has a worst-case regret bound*

$$
\begin{aligned}
\sup_{M \in \mathcal{M}} \; &\text{Regret}(M, K, N, \pi, \text{RLSVI}_{\beta,\alpha,\xi}) \\
&\leq \widetilde{O}(\epsilon KHN + KH^{5/2}\Gamma\sqrt{N}),
\end{aligned}
\tag{9}
$$

*where $\widetilde{O}(\cdot)$ hides the dependence on logarithmic factors.*

Since the only difference between the algorithms is the amount of data stored, and their structures are identical, we only prove (9), with (8) following immediately by reducing a factor of $\sqrt{K}$ from the regret bound in (9). We omit the redundant proof for Algorithm 1 and defer the proof for (9) to Appendix C.

**Comparison with worst-case regret bounds from (Russo, 2019; Agrawal et al., 2021)** A worst-case regret bound of $\tilde{O}(H^3 S^{3/2}\sqrt{AK})$ was obtained in (Russo, 2019) for a single-agent version of RLSVI algorithm. This bound was improved later by (Agrawal et al., 2021) to $\tilde{O}(H^{5/2}S\sqrt{AK})$. For the single-agent case with $N = 1$, Algorithm 1 results in a worst-case regret bound of $\tilde{O}(\sqrt{K}H^{5/2}\Gamma)$, which translates into $\tilde{O}(H^2\Gamma\sqrt{T})$. So our bound matches that of (Agrawal et al., 2021) if $\Gamma = S \times A$ and $S \approx \sqrt{\Gamma}$. Algorithm 3 implies a worst-case regret bound of $\tilde{O}(KH^{5/2}\Gamma)$. Here an extra $\sqrt{K}$ compared to that of (Agrawal et al., 2021; Russo, 2019) comes from the fact we only keep the trajectories of the agents from the previous episode rather than all the episodes up to the current period as in (Russo, 2019; Agrawal et al., 2021) to reduce space complexity. The extra $\epsilon$ term for both algorithms comes from model misspecification of state-aggregation formulation, which is similar to the result in (Dong et al., 2019b).

At each time period $h$, each agent gets $N$ samples of tuples $(s, a, r, s')$, and they share information by the aggregation of information through the computation of a weighted $Q$-value (6) at the end of each episode. With this trick, though agents learn their own policies concurrently in parallel within each episode, we are able to obtain a sub-linear worst-case regret bound of $O(\sqrt{N})$ with respect to the number of agents.

## 4. Infinite-horizon Concurrent Learning

We now turn to the infinite-horizon case. Consider an unknown fixed environment as $M = (T, \mathcal{A}, \mathcal{S}, P, r, N)$, with $N$ agents interacting in $M$. Here $\mathcal{A} = [A]$ is the action space, $\mathcal{S} = [S]$ is the state space, $P(s' \mid s, a)$ is the transition probability from $s \in \mathcal{S}$ to $s' \in \mathcal{S}$ given action $a \in \mathcal{A}$. After agent $p$ selects action $a_t^p$ at state $s_t^p$, the agent observes $s_{t+1}^p$ and receives a fixed reward $r_{t+1}^p = r(s_t^p, a_t^p)$ where $r \in [0, 1]$. A stochastic policy $\pi$ can be represented by a probability mass function $\pi(\cdot|s_t)$ that an agent assigns to actions in $\mathcal{A}$ given situation state $s_t$. For a policy $\pi$, we denote the average reward starting at state $s$ as

$$
\lambda_\pi(s) = \liminf_{T \to \infty} \mathbb{E}\left[\frac{1}{T}\sum_{t=0}^{T-1} r_{t+1}\Big| s_1 = s\right].
\tag{10}
$$

For any state $s \in \mathcal{S}$, denote the optimal average reward as $\lambda_*(s) = \sup_\pi \lambda_\pi(s)$. We consider weakly-communicating MDP, which is defined as follows:

**Definition 3** (Weakly-communicating MDP). *A MDP is weakly communicating if there exists a closed subset of states, where each state within is reachable from any other state within that set under some deterministic stationary reward. And there exists a transient subset of states (possibly empty) under every policy.*

For any $s, s' \in \mathcal{S}$ and $a \in \mathcal{A}$, denote $P_{s,a,s'} = P(s'|s,a)$. For each policy $\pi$ define transition probabilities under $\pi$ as $P_{s,\pi,s'} = \sum_{a \in \mathcal{A}} \pi(a|s) P_{s,a,s'}$, and reward as $r_{s,\pi} = \sum_{a \in \mathcal{A}} \pi(a|s) r(s,a)$.

**Pseudo-episodes** We extend our concurrent learning framework for the finite-horizon case to the infinite-horizon case by incorporating the idea of pseudo-episodes from (Xu et al., 2022). Suppose time step $t$ is the beginning of a pseudo-episode when we sample a random variable $H \sim \text{Geometric}(1 - \eta)$, where $\text{Geometric}(1 - \eta)$ is geometric distribution with parameter $1 - \eta$. In the numerical experiment (section 6), we set $\eta = 0.99$. The agents compute new policies respectively according to the collected trajectories from last pseudo-episode, and sample their own MDPs respectively at time steps $t + 1, \ldots, t + H - 1$. Now the beginning of the next pseudo-episode is set as $t + H$. We use $\mathcal{H}_{t_1,t_2} = \bigcup_{p=1}^{N} \bigcup_{i=t_1}^{t_2} \{s_i^p, a_i^p, r_i^p\}$ to denote the trajectories of all agents from time step $t_1$ to time step $t_2$. For policy $\pi$, denote the $\eta$-discounted value function as $V_\pi^\eta \in \mathbb{R}^S$, then we have

$$V_\pi^\eta := \mathbb{E}_H \left[ \sum_{h=0}^{H-1} P_\pi^h r_\pi \Big| M \right] = \mathbb{E} \left[ \sum_{h=0}^{\infty} \eta^h P_\pi^h r_\pi \Big| M \right],$$
(11)

where the expectation is taken over the random episode length $H$. A policy is said to be optimal if $V_\pi^\eta = \sup_{\pi'} V_{\pi'}^\eta$. For an optimal policy, we also write $V_*^\eta(s) \equiv V_\pi^\eta(s)$ as the optimal value. Note that $V_\pi^\eta \in \mathbb{R}^S$ satisfies the Bellman equation $V_\pi^\eta = r_\pi + \eta P_\pi V_\pi^\eta$. For any $(s,a)$, define $Q_\pi^\eta(s,a) = r(s,a) + \eta P V_\pi^\eta(s)$, where we use the notation that $P_\pi V(s) = \mathbb{E}_{s' \sim P_{s,\pi(s)}}[V(s')]$. Correspondingly, define

$$Q_*^\eta(s,a) = r(s,a) + \eta P V_*^\eta(s').$$

By definition, we have $V_*^\eta(s) = \max_{a \in \mathcal{A}} Q_*^\eta(s,a)$.

**Discounted Regret** To analyze the algorithm over $T$ time steps, consider $K = \arg\max\{k : t_k \leq T\}$ as the number of pesudo-episodes until time $T$. We use the convention that $t_{K+1} = T+1$. Given a discount factor $\eta \in [0, 1)$, define the $\eta$-discounted regret up to time $T$ as $\text{Regret}_\eta(M, T, \pi) = \sum_{k=1}^{K} \Delta_k$, where $\Delta_k$ is the total regret of all $N$ agents over pseudo-episode $k$: $\Delta_k = \sum_{p=1}^{N} V_*^\eta(s_{k,1}^p) - V_{\pi^{kp}}^\eta(s_{k,1}^p)$, where $V_*^\eta = V_{\pi^*}^\beta$, policy $\pi^{kp}$ is computed from the trajectory $\mathcal{H}_{t_{k-1}, t_k - 1}$ from pseudo-episode $k - 1$ by agent $p$, and $a_t^p \sim \pi^{kp}(\cdot \mid s_t^p), s_{t+1}^p \sim P(\cdot|s_t^p, a_t^p), r_t^p = r(s_t^p, a_t^p)$

for $t \in E_k$, and $E_k$ denotes the time steps within pseudo-episode $k$. So the discounted regret is a random variable depending on the algorithm's random sampling, and the random lengths of the pseudo-episodes, and as a result,

$$\Delta_k = \mathbb{E}_{H_k} \left[ \sum_{p=1}^{N} \sum_{h=0}^{H_k} (P_{\pi^*}^h r_{\pi^*} - P_{\pi^{kp}} r_{\pi^{kp}}) \mid M \right]$$
$$= \mathbb{E} \left[ \sum_{p=1}^{N} \sum_{h=0}^{\infty} \eta^h (P_{\pi^*}^h r_{\pi^*} - P_{\pi^{kp}} r_{\pi^{kp}}) \mid M \right].$$

**Regret** The optimal average reward $\lambda_*$ is state-independent under a weakly-communicating MDP. The agent $p$ selects a policy $\pi^{kp}$ and executes it within the $k^{th}$ pseudo-episode. The cumulative expected regret incurred by the collection of policies $\pi = \{\pi^{kp}\}_{k \in [K], p \in [N]}$ over $T$ time steps and across $N$ agents with the fixed environment $M$ is

$$\text{Regret}(M, T, N, \pi) := \mathbb{E}_K \left[ \sum_{k=1}^{K} \Delta_k \Big| M \right], \quad (12)$$

where the expectation is taken over the random seeds used by the randomized algorithm, conditioning on the true MDP $M$. In the following, we denote $(s_{k,h}^p, a_{k,h}^p, r_{k,h}^p)$ as the state, action and reward for agent $p$ during pseudo-episode $k$ and period $h$.

**Empirical Estimation** We let $n_k(s,a)$ be the total number of times that $(s,a)$-pair appears during the $k^{th}$ pseudo-episode, such that

$$n_k(s,a) = \sum_{p=1}^{N} \sum_{t=t_k}^{t_{k+1}-1} \mathbf{1} \{(s_t^p, a_t^p) = (s,a)\}.$$

Then $\forall s'$, the empirical estimate $\hat{P}(s'|s,a)$ of the transition probability for pseudo-episode $k$ is

$$\hat{P}_k(s'|s,a)$$
$$= \frac{\sum_{p=1}^{N} \sum_{t \in E_{k-1}} \mathbf{1} \{(s_t^p, a_t^p, s_{t+1}^p) = (s,a,s')\}}{n_{k-1}(s,a)}.$$
(13)

The empirical estimate of the corresponding reward is

$$\hat{R}_k(s,a)$$
$$= \frac{\sum_{p=1}^{N} \mathbf{1}\{(s_t^p, a_t^p) = (s,a)\} \sum_{t \in E_{k-1}} r(s_t^p, a_t^p)}{n_{k-1}(s,a)}.$$
(14)

For the second empirical estimation method, we utilize the full historical data, and similar to the finite-horizon case, the empirical estimates for transition probability and reward are

$$\hat{P}_k^{\texttt{full}}(s'|s,a)$$
$$= \frac{\sum_{i=0}^{k-1} \sum_{p=1}^{N} \sum_{t \in E_i} \mathbf{1} \{(s_t^p, a_t^p, s_{t+1}^p) = (s,a,s')\}}{\sum_{i=0}^{k-1} n_i(s,a)}.$$

$$\hat{R}_k^{\texttt{full}}(s,a)$$
$$= \frac{\sum_{p=1}^{N} \sum_{i=0}^{k-1} \mathbf{1}\{(s_t^p, a_t^p) = (s,a)\} \sum_{t \in E_i} r(s_t^p, a_t^p)}{\sum_{i=0}^{k-1} n_i(s,a)}.$$

**Aggregated States**  We extend the aggregated states in the finite-horizon case to the infinite horizon case. Let $\Phi$ be the set of all aggregated states, and let $\phi : \mathcal{S} \times \mathcal{A} \to \Phi$ be the mapping from state-action pairs to aggregated states. We let $\Phi = [\Gamma]$. The aggregated representation for the infinite-horizon case is defined as follows:

**Definition 4.** *We say that $\phi$ is an $\epsilon$-error aggregated state-representation (or $\epsilon$-error aggregation) of an MDP, if for all $s, s' \in \mathcal{S}$, $a, a' \in \mathcal{A}$ such that $\phi(s,a) = \phi(s',a')$, we have $|Q_*^{\eta}(s,a) - Q_*^{\eta}(s',a')| \le \epsilon$.*

We are now ready to present the concurrent learning algorithm for the infinite-horizon case and the theoretical guarantee, as detailed in the next section.

## 5. Infinite-Horizon Algorithm and Regret Bound

The concurrent learning algorithm for the infinite-horizon MDP is summarized as Algorithm 4. Our result is based on the following definition of reward averaging time proposed by (Dong et al., 2022).

**Definition 5.** *The reward averaging time $\tau_\pi$ of a policy $\pi$ is the smallest value $\tau \in [0, \infty)$ such that $\forall T \ge 0, s \in \mathcal{S}$, $\left| \mathbb{E}_\pi \left[ \sum_{t=0}^{T-1} r_{t+1} \Big| s_0 = s \right] - T \cdot \lambda_\pi(s) \right| \le \tau$.*

Typically the regret bounds established in the literature requires assessing an optimal policy within bounded time. Examples include episode duration (Osband et al., 2013; Jin et al., 2018), diameter (Auer et al., 2008), or span (Bartlett & Tewari, 2012). Policies that require intractably large amount of time are infeasible in practice. So we impose the following assumption:

**Assumption 6.** *For any weakly communicating MDP $M$ with state space $\mathcal{S}$ and action space $\mathcal{A}$, $\exists \tau < \infty$ such that $\tau_* \le \tau$.*

When $\pi^*$ is an optimal policy for $M$, $\tau_* := \tau_{\pi^*}$ is equivalent to the notion of span in (Bartlett & Tewari, 2012). Let $\mathcal{M}$ be the set of infinite-horizon weakly-communicating MDPs with state space size $S$, action space size $A$, rewards bounded in $[0, 1]$ that satisfy Assumption 6. Let $N$ be the number of agents interacting in the same environment. Recall that $\tau$ is given by Assumption 6.

Suppose $\{\phi\}$ is an $\epsilon$-error aggregation (defined as in Definition 4) of the underlying MDP. For a tuning parameter sequences $\beta = \{\beta_n\}_{n \in \mathbb{N}}, \alpha = \{\alpha_n\}_{n \in \mathbb{N}}, \xi = \{\xi_n\}_{n \in \mathbb{N}}$, where for $k$ as the index of pseudo-episode,

$\beta_k = \frac{1}{2}\tau^3 \log(2\tau\Gamma k)$, $\alpha_n = \frac{1}{1+n}$, and

$$\begin{aligned} \xi_n &= \epsilon + \frac{2\alpha_n \sqrt{\log(2TN/\delta)}}{(1-\eta)\sqrt{\max\{n,1\}}} \\ &\quad + \frac{2\alpha_n \sqrt{\beta_{k-1} \log(2TN/\delta)}}{\sqrt{(n+1)\max\{n,1\}}} \end{aligned} \tag{15}$$

**Theorem 7** (Infinite-horizon Worst-case Regret Bound).
*Algorithm 2 has a worst-case regret bound*

$$\begin{aligned} &\sup_{M \in \mathcal{M}} \text{Regret}(M, T, N, RLSVI_{\beta,\alpha,\xi}^{full}) \\ &\le \widetilde{O}(\epsilon\sqrt{T}N + \tau^{3/2}\Gamma\sqrt{TN}). \end{aligned} \tag{16}$$

*Algorithm 4 has a worst-case regret bound*

$$\begin{aligned} &\sup_{M \in \mathcal{M}} \text{Regret}(M, T, N, RLSVI_{\beta,\alpha,\xi}) \\ &\le \widetilde{O}(\epsilon TN + \tau^{3/2}T\Gamma\sqrt{N}). \end{aligned} \tag{17}$$

Note that the bound in (17) matches that of the finite-horizion case (9) by noting that $T = KH$ for the finite-horizon case. And $\tau^{3/2}$ corresponds to the $H^{3/2}$ factor in the finite-horizon case bound by noting that taking $\tau = H$ makes the condition holds in Definition 5 in the finite-horizon case with $T = KH$. Similar to the finite-horizon case, (16) follows directly from (17) with a $\sqrt{T}$ reduction, as Algorithm 2 utilizes the full history, whereas Algorithm 4 retains only the last pseudo-episode. The intuition and worst-case regret comparison follow the discussion after Theorem 2.

## 6. Numerical Experiments

We present numerical results for both finite-horizon and infinite-horizon cases in Figure 1. For the finite-horizon case $(S, A, K, H)$ or the infinite-horizon case $(S, A, T)$, the transition probabilities are drawn from a Dirichlet distribution, and rewards, fixed as deterministic, are uniformly distributed on $[0, 1]$, forming inherent features of the MDP class. The finite-horizon case settings are (i) $K = 20, H = 30, S = 5, A = 5$; (ii) $K = 25, H = 40, S = 10, A = 10$; (iii) $K = 30, H = 50, S = 20, A = 20$. The infinite-horizon case settings are with $T = 300$ and (i) $S = 5, A = 5$; (ii) $S = 20, A = 20$; (iii) $S = 30, A = 30$, where $\eta = 0.99$ in the pseudo-episode sampling. Under each setting, we compare the results for $N = 1, 3, 5, 7, 10, 15, 20, 30, 40, 50$, with $\epsilon = 0$. We set $\eta = 0.99$.

For each agent number $N$ in the finite-horizon setting, we sample 500 MDPs from the defined class. For each sampled MDP, we compute the cumulative regret over time and then identify the maximum regret across all 500 instances, representing the worst-case regret in our analysis.

For the infinite-horizon setting, we estimate regret by averaging over 50 geometric segmentations of $[T]$ per MDP,

consistent with the definition of infinite-horizon regret based on pseudo-episodes in (12). The worst-case regret is then obtained by taking the maximum across 500 sampled MDPs.

Figure 1 illustrates a $1/\sqrt{N}$ decreasing trend in per-agent regret for both settings, consistent with our theoretical predictions. The replication code is available at `https://github.com/yz2/rlsvi_code`.

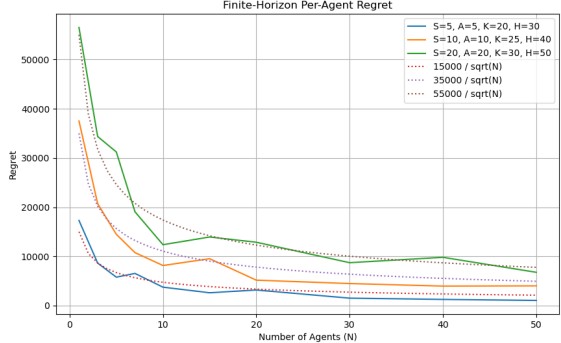

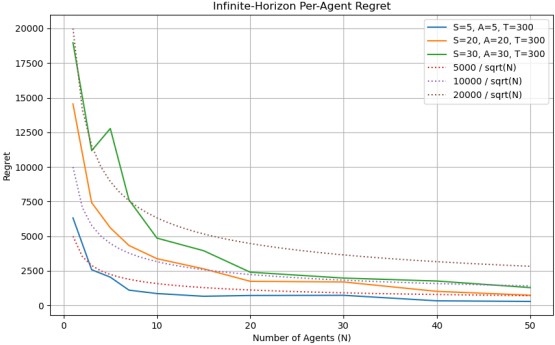

*Figure 1.* Per-agent regret vs number of agents for finite-horizon (top panel) and infinite-horizon (bottom panel) settings. The solid curves represent the per-agent worst-case regret computed from 500 random environments. The dashed ones are the reference curves of $\mathrm{constant}/\sqrt{N}$ fitting the $\Theta(1/\sqrt{N})$ trend as we show in our theoretical results.

## Acknowledgements

We gratefully acknowledge the support from the NSF grant CCF-2312205 and the ONR grant ONR 13983263. We would like to thank Pratap Ranade and Engin Ural for inspiring discussions on society of agents that have helped shape and push an ambitious vision of this research agenda, for which this work is only an initial step.

## Impact Statement

This paper presents work whose goal is to advance the field of Machine Learning. There are many potential societal consequences of our work, none which we feel must be specifically highlighted here.

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

## A. Comparison Table: Existing Work vs. Our Approach

*Table 1.* Comparison of regret bounds for various RLSVI/LSVI algorithms

| Agent | Setup | Algorithm | Regret Bound | Regret-Type | Data Stored | Numerical |
|---|---|---|---|---|---|---|
| Single | Tabular | RLSVI (Russo, 2019) | $\tilde{O}\big(H^3 S^{3/2}\sqrt{AK}\big)$ | Worst-case | All-history | N/A |
| Single | Tabular | RLSVI (Agrawal et al., 2021) | $\tilde{O}\big(H^{5/2} S\sqrt{AK}\big)$ | Worst-case | All-history | N/A |
| Multi | Tabular | Concurrent RLSVI (Taiga et al., 2022) | N/A | Bayes | All-history | Synthetic |
| Multi | Linear Functional Approximation | Concurrent LSVI (Desai et al., 2018) | $\tilde{O}\big(H^2 \sqrt{d^3 KN}\big)$ | Worst-case | All-history | N/A |
| Multi | Linear Functional Approximation | Concurrent LSVI (Min et al., 2023) | $\tilde{O}\big(H\sqrt{dKN}\big)$ | Worst-case | All-history | N/A |
| Multi | Tabular | Concurrent RLSVI (ours-1) | $\tilde{O}\big(H^{5/2}\Gamma\sqrt{KN}\big)$ | Worst-case | All-history | N/A |
| Multi | Tabular | Concurrent RLSVI (ours-2) | $\tilde{O}\big(H^{5/2}\Gamma K\sqrt{N}\big)$ | Worst-case | One episode | Synthetic |

We outline several key observations for Table 1 below:

- The methods marked as "N/A" in the "numerical" column store all agents' trajectories at every step, making them computationally infeasible as $N$ grows large. These approaches (including our first finite-horizon algorithm, second-to-last row in the table) provide only theoretical analyses without numerical results.

- Although Taiga et al. (2022) stores all data empirically, their RLSVI algorithm assumes a known parametric feature matrix, making it simpler to implement than ours. Also they evaluate only Bayes regret—less stringent than our worst-case regret—and provide empirical results exclusively on synthetic data without theoretical guarantees.

- The last row corresponds to storing only the latest episode data, increasing the regret bound by a factor of $\sqrt{K}$ but reduces space complexity by a factor of $K$, making it computationally feasible.

## B. Algorithms

Algorithm 1 and Algorithm 3 are finite-horizon algorithms, where the former stores all historical data, and the latter only keeps one episode in the buffer. Algorithm 2 and Algorithm 4 are infinite-horizon algorithms, where the former stores all historical data and the latter only keeps one pseudo-episode.

---

**Algorithm 1** Concurrent RLSVI: Finite-Horizon Storing All Historical Data

---

**Data:** $K, H, S, \mathcal{A}, N, \mathbf{s}_1, \{\phi_h\}_{h=1}^H$, Tuning parameters $\{\beta_k\}_{k \in \mathbb{N}}$

Define constants $\alpha_t \leftarrow 1/(1+t), t = 1, 2, \ldots$

/* Define squared temporal difference error */

$\mathcal{L}(Q \mid Q_{k-1,h}, Q_{\text{next}}, \mathcal{D}, \xi, \alpha) = \sum_{(s,a,r,s') \in \mathcal{D}} (Q - \xi - (1-\alpha)Q_{k-1,h}(\phi_h(s,a)) - \alpha(r + \max_{a' \in \mathcal{A}} Q_{\text{next}}(s',a')))^2$

**Initialize:**

$\hat{Q}_{0h}^p(\gamma) = H, \forall h \in [H], \gamma \in [\Gamma], p \in [N]$

Each agent randomly samples the initial trajectory $\{s_{0,1}^p, a_{0,1}^p, r_{0,1}^p, \ldots, s_{0,H}^p, a_{0,H}^p, r_{0,1}^H\}_{p=1}^N$, with $s_{0,1}^p = s_1^p$

$N_{0,h}(\gamma) = \sum_{p=1}^N \mathbf{1}\{\phi_h(s_{0,h}^p, a_{0,h}^p) = \gamma\}, \forall \gamma \in [\Gamma], h \in [H]$

compute $\hat{Q}_{0,h}$ by ( 6)

**for** *episode* $k = 1, 2, \ldots$ **do**

   /*Each agent rollouts in the evironment*/

   **for** $p = 1, \ldots, N$ **do**

      /*Executed in parallel*/

      **for** *period* $h = 1, \ldots, H$ **do**

         $a_{k,h}^p \leftarrow \arg\max_{a \in \mathcal{A}} \hat{Q}_{k-1,h}^p(\phi_h(s_{k,h}^p, a))$

         observe reward $r_{k,h}^p$ and next state $s_{k,h+1}^p$

         $\mathcal{D}_h \leftarrow \mathcal{D}_h \cup \{(s_{k,h}^p, a_{k,h}^p, r_{k,h}^p, s_{k,h+1}^p)\}$

      **end**

   **end**

   /*Visitation of aggregated-states*/

   $N_{k,h}(\gamma) \leftarrow \sum_{i=1}^{k-1} \sum_{p=1}^N \mathbf{1}\{\phi_h(s_{i,h}^p, a_{i,h}^p) = \gamma\}, \forall \gamma \in [\Gamma], h \in [H]$

   /*Construct perturbed data sets and sample regularization noise $\tilde{Q}$*/

   **for** $p \in [N]$ *and* $h \in [H]$ **do**

      /*Executed in parallel*/

      For any $(s,a) \in \mathcal{S} \times \mathcal{A}$, sample array

      $\tilde{Q}_{kh}^p(s,a) \sim \mathcal{N}\left(0, \frac{\beta_k}{1 + N_{k,h}(\phi_h(s,a))}\right)$,

      /* Draw prior sample */

      $\tilde{D}_{kh}^p \leftarrow \{\}$

      **for** $(s,a,r,s') \in \mathcal{D}_h^k$ **do**

         Sample $w^p(s,a) \sim \mathcal{N}\left(0, \frac{\beta_k}{1 + N_{k,h}(\phi_h(s,a))}\right)$

         $\tilde{D}_{kh}^p \leftarrow \tilde{D}_{kh}^p \cup \{(s,a,r+w^p,s')\}$

      **end**

   **end**

   /*Estimate Q on perturbed data*/

   **for** $p = 1, \ldots, N$ **do**

      /*Executed in parallel*/

      Define terminal value $\hat{Q}_{k,H+1}^p(\gamma) \leftarrow H \ \forall \gamma \in [\Gamma]$

      **for** *period* $h = H, \ldots, 1$ **do**

         $\bar{Q}_{k,h}^p(\gamma) \leftarrow \arg\min_{Q \in \mathbb{R}} \mathcal{L}(Q | \hat{Q}_{k-1,h}, \hat{Q}_{k,h+1}^p, \tilde{D}_{kh}^p, \xi, \alpha) + \|Q - \alpha_{N_{k-1,h}(\gamma)} \tilde{Q}_{kh}^p\|_2^2, \forall \gamma \in [\Gamma]$

         $\hat{Q}_{k,h}^p(\gamma) \leftarrow \min\{\bar{Q}_{k,h}^p(\gamma), H\}, \forall \gamma \in [\Gamma]$

      **end**

      $s_{k,1}^p \leftarrow s_1^p$

   **end**

   Update $\hat{Q}_{k,h}(\gamma), \forall \gamma \in [\Gamma]$ by ( 6)

**end**

---

---

**Algorithm 2** Concurrent RLSVI: Infinite-Horizon Storing All Historical Data

---

**Data:** Discount factor $\eta$, $t_0 = 1$, $t = 1$, $k = 0$, $X_1 = 0$, $S, A, N, T, \phi$, tuning parameters $\{\beta_k\}_{k \in \mathbb{N}}, \xi, \eta$

**Initialize** $N_k(\gamma) \leftarrow 0, \forall \gamma \in [\Gamma], k \in [K]; \hat{Q}_0(\gamma) \leftarrow 0, \forall \gamma \in [\Gamma]$

Define constants $\alpha_t \leftarrow 1/(1+t), t = 1, 2, \dots$

/* Define squared temporal difference error */

$\mathcal{L}(Q \mid \hat{Q}, Q_{\text{next}}, \mathcal{D}, \xi, \eta, \alpha) = \sum_{(s,a,r,s') \in \mathcal{D}} (Q - \eta\xi - \eta(1-\alpha)\hat{Q}(\phi(s,a)) - \alpha\eta(r + \max_{a' \in \mathcal{A}} Q_{\text{next}}(s', a')))^2$

Sample $H_0 \sim \text{Geometric}(1 - \eta)$, set $H_0 \leftarrow \min\{H_0, T + 1 - t\}$

Each agent randomly samples the initial trajectory $\{s_{0,1}^p, a_{0,1}^p, r_{0,1}^p, \dots, s_{0,H}^p, a_{0,H}^p, r_{0,H}^p\}_{p=1}^N$, with $s_{0,1}^p = s_1^p$

$k \leftarrow k + 1$, $t_k = 1 + H_0$

$t_k \leftarrow$ the start time of pseudo-episode $k$

**while** $t \leq T$ **do**

  Sample $H \sim \text{Geometric}(1 - \eta)$

  $H \leftarrow \min\{H, T + 1 - t\}$

  $t_{k+1} \leftarrow t_k + H$ (the start time of pseudo-episode $k + 1$)

  /*Each agent rollouts in the environment */

  **for** $p = 1, \dots, N$ **do**

    /*Executed in parallel*/

    **for** $t = t_k, \dots, t_{k+1} - 1$ **do**

      $a_t^p \leftarrow \arg\max_{a \in \mathcal{A}} \hat{Q}_t^p(\phi(s_t^p, a))$

      observe reward $r_t^p$ and next state $s_{t+1}^p$

      $\mathcal{D}_k \leftarrow \mathcal{D}_k \cup \{(s_t^p, a_t^p, r_t^p, s_{t+1}^p)\}$

    **end**

    /* Visitation of aggregated-states */

    $N_{k-1}(\gamma) \leftarrow \sum_{p=1}^N \sum_{t=0}^{t_k - 1} \mathbf{1}\{\phi(s_t^p, a_t^p) = \gamma\}, \forall \gamma \in [\Gamma]$

  **end**

  /* Construct perturbed datasets and sample regularization noise $\tilde{Q}$ */

  **for** $p \in [N]$ *and* $t = t_k, \dots, t_{k+1} - 1$ **do**

    /* Executed in parallel */

    Sample array $\tilde{Q}_t^p(s, a) \sim \mathcal{N}(0, \frac{\beta_{N_{k-1}(\phi(s,a))}}{N_{k-1}(\phi(s,a)) + 1}), \forall (s, a)$

    /* Draw prior sample */

    $\mathcal{H}_k^p \leftarrow \{\}$

    **for** $(s, a, r, s') \in \mathcal{D}_k$ **do**

      /* Randomly perturb data */

      Sample $w^p(s, a) \sim \mathcal{N}(0, \frac{\beta_{N_{k-1}(\phi(s,a))}}{N_{k-1}(\phi(s,a)) + 1})$

      $\mathcal{H}_k^p \leftarrow \mathcal{H}_k^p \cup \{(s, a, r + w^p, s')\}$

    **end**

  **end**

  /* Estimate $Q$ on perturbed data */

  **for** $p = 1, \dots, N$ **do**

    /* Executed in Parallel */

    Define terminal value $\hat{Q}_{t_{k+1}}^p(\gamma) \leftarrow 0 \ \forall \gamma \in [\Gamma]$

    **for** $t = t_{k+1} - 1, \dots, t_k$ **do**

      /* Estimate Q on noisy data */

      $\hat{Q}_t^p \leftarrow \arg\min_{Q \in \mathbb{R}} \mathcal{L}(Q \mid \hat{Q}_{k-1}, \hat{Q}_{t+1}^p, \mathcal{H}_k^p, \xi, \eta, \alpha_{N_{k-1}(\gamma)}) + \|Q - \eta\alpha_{N_{k-1}(\gamma)}\tilde{Q}_t^p\|_2^2$

      $\hat{Q}_t^p \leftarrow \min\{\hat{Q}_t^p, \frac{1}{1-\eta}\}, \forall \gamma \in [\Gamma]$

    **end**

    $s_{t_k}^p = s_1^p, \forall p \in [N]$.

  **end**

  $\hat{Q}_k(\gamma) = \frac{\sum_{p=1}^N \sum_{t=t_k}^{t_{k+1} - 1} \mathbf{1}\{\phi(s_t^p, a_t^p) = \gamma\} \hat{Q}_{t_k}^p(\gamma)}{N_k(\gamma)}, \forall \gamma \in \Gamma$

  $t \leftarrow t_{k+1}$, $k \leftarrow k + 1$

**end**

---

---

**Algorithm 3** Concurrent RLSVI: Finite-Horizon (Storing the Data of One Episode)

---

**Data:** $K, H, S, \mathcal{A}, N, \mathbf{s}_1, \{\phi_h\}_{h=1}^H$, Tuning parameters $\{\beta_k\}_{k \in \mathbb{N}}$

Define constants $\alpha_t \leftarrow 1/(1+t), t = 1, 2, \ldots$

/* Define squared temporal difference error */

$\mathcal{L}(Q \mid Q_{k-1,h}, Q_{\text{next}}, \mathcal{D}, \xi, \alpha) = \sum_{(s,a,r,s') \in \mathcal{D}} (Q - \xi - (1-\alpha)Q_{k-1,h}(\phi_h(s,a)) - \alpha(r + \max_{a' \in \mathcal{A}} Q_{\text{next}}(s', a')))^2$

**Initialize:**

$\hat{Q}_{0h}^p(\gamma) = H, \forall h \in [H], \gamma \in [\Gamma], p \in [N]$

Each agent randomly samples the initial trajectory $\{s_{0,1}^p, a_{0,1}^p, r_{0,1}^p, \ldots, s_{0,H}^p, a_{0,H}^p, r_{0,1}^H\}_{p=1}^N$, with $s_{0,1}^p = s_1^p$

$N_{0,h}(\gamma) = \sum_{p=1}^N \mathbf{1}\{\phi_h(s_{0,h}^p, a_{0,h}^p) = \gamma\}, \forall \gamma \in [\Gamma], h \in [H]$

compute $\hat{Q}_{0,h}$ by ( 6)

**for** *episode* $k = 1, 2, \ldots$ **do**
  /*Each agent rollouts in the evironment*/
  **for** $p = 1, \ldots, N$ **do**
    /*Executed in parallel*/

    **for** *period* $h = 1, \ldots, H$ **do**
      $a_{k,h}^p \leftarrow \arg\max_{a \in \mathcal{A}} \hat{Q}_{k-1,h}^p(\phi_h(s_{k,h}^p, a))$
      observe reward $r_{k,h}^p$ and next state $s_{k,h+1}^p$
      $\mathcal{D}_h^k \leftarrow \mathcal{D}_h^k \cup \{(s_{k,h}^p, a_{k,h}^p, r_{k,h}^p, s_{k,h+1}^p)\}$
    **end**
  **end**
  /*Visitation of aggregated-states*/
  $N_{k,h}(\gamma) \leftarrow \sum_{p=1}^N \mathbf{1}\{\phi_h(s_{k,h}^p, a_{k,h}^p) = \gamma\}, \forall \gamma \in [\Gamma], h \in [H]$
  /*Construct perturbed data sets and sample regularization noise $\tilde{Q}$*/
  **for** $p \in [N]$ *and* $h \in [H]$ **do**
    /*Executed in parallel*/
    For any $(s, a) \in \mathcal{S} \times \mathcal{A}$, sample array
    $\tilde{Q}_{kh}^p(s, a) \sim \mathcal{N}\left(0, \frac{\beta_k}{1 + N_{k,h}(\phi_h(s,a))}\right),$
    /* Draw prior sample */
    $\tilde{D}_{kh}^p \leftarrow \{\}$
    **for** $(s, a, r, s') \in \mathcal{D}_h^k$ **do**
      Sample $w^p(s, a) \sim \mathcal{N}\left(0, \frac{\beta_k}{1 + N_{k,h}(\phi_h(s,a))}\right)$
      $\tilde{D}_{kh}^p \leftarrow \tilde{D}_{kh}^p \cup \{(s, a, r + w^p, s')\}$
    **end**
  **end**
  /*Estimate Q on perturbed data*/
  **for** $p = 1, \ldots, N$ **do**
    /*Executed in parallel*/
    Define terminal value $\hat{Q}_{k,H+1}^p(\gamma) \leftarrow H \ \forall \gamma \in [\Gamma]$

    **for** *period* $h = H, \ldots, 1$ **do**
      $\bar{Q}_{k,h}^p(\gamma) \leftarrow \arg\min_{Q \in \mathbb{R}} \mathcal{L}(Q | \hat{Q}_{k-1,h}, \hat{Q}_{k,h+1}^p, \tilde{D}_{kh}^p, \xi, \alpha) + \|Q - \alpha_{N_{k-1,h}(\gamma)} \tilde{Q}_{kh}^p\|_2^2, \forall \gamma \in [\Gamma]$
      $\hat{Q}_{k,h}^p(\gamma) \leftarrow \min\{\bar{Q}_{k,h}^p(\gamma), H\}, \forall \gamma \in [\Gamma]$
    **end**
    $s_{k,1}^p \leftarrow s_1^p, \mathcal{D}_h^k = \emptyset$

  **end**
  Update $\hat{Q}_{k,h}(\gamma), \ \forall \gamma \in [\Gamma]$ by ( 6)

**end**

---

---

**Algorithm 4** Concurrent RLSVI: Infinite-Horizon (Storing the Data of One Pseudo-episode)

---

**Data:** Discount factor $\eta$, $t_0 = 1$, $t = 1$, $k = 0$, $X_1 = 0$, $S, A, N, T, \phi$, tuning parameters $\{\beta_k\}_{k \in \mathbb{N}}, \xi, \eta$

**Initialize** $N_k(\gamma) \leftarrow 0, \forall \gamma \in [\Gamma], k \in [K]$; $\hat{Q}_0(\gamma) \leftarrow 0, \forall \gamma \in [\Gamma]$

  Define constants $\alpha_t \leftarrow 1/(1+t)$, $t = 1, 2, \dots$

  /* Define squared temporal difference error */

  $\mathcal{L}(Q \mid \hat{Q}, Q_{\text{next}}, \mathcal{D}, \xi, \eta, \alpha) = \sum_{(s,a,r,s') \in \mathcal{D}} (Q - \eta \xi - \eta(1-\alpha)\hat{Q}(\phi(s,a)) - \alpha \eta (r + \max_{a' \in \mathcal{A}} Q_{\text{next}}(s', a')))^2$

  Sample $H_0 \sim \text{Geometric}(1 - \eta)$, set $H_0 \leftarrow \min\{H_0, T + 1 - t\}$

  Each agent randomly samples the initial trajectory $\{s_{0,1}^p, a_{0,1}^p, r_{0,1}^p, \dots, s_{0,H}^p, a_{0,H}^p, r_{0,H}^p\}_{p=1}^N$, with $s_{0,1}^p = s_1^p$

  $k \leftarrow k + 1$, $t_k = 1 + H_0$

  $t_k \leftarrow$ the start time of pseudo-episode $k$

  **while** $t \leq T$ **do**

    Sample $H \sim \text{Geometric}(1 - \eta)$

    $H \leftarrow \min\{H, T + 1 - t\}$

    $t_{k+1} \leftarrow t_k + H$ (the start time of pseudo-episode $k+1$)

    /*Each agent rollouts in the environment */

    **for** $p = 1, \dots, N$ **do**

      /*Executed in parallel*/

      **for** $t = t_k, \dots, t_{k+1} - 1$ **do**

        $a_t^p \leftarrow \arg\max_{a \in \mathcal{A}} \hat{Q}_t^p(\phi(s_t^p, a))$

        observe reward $r_t^p$ and next state $s_{t+1}^p$

        $\mathcal{D}_k \leftarrow \mathcal{D}_k \cup \{(s_t^p, a_t^p, r_t^p, s_{t+1}^p)\}$

      **end**

      /* Visitation of aggregated-states */

      $N_{k-1}(\gamma) \leftarrow \sum_{p=1}^N \sum_{t=t_{k-1}}^{t_k - 1} \mathbf{1}\{\phi(s_t^p, a_t^p) = \gamma\}, \ \forall \gamma \in [\Gamma]$

    **end**

    /* Construct perturbed datasets and sample regularization noise $\tilde{Q}$ */

    **for** $p \in [N]$ *and* $t = t_k, \dots, t_{k+1} - 1$ **do**

      /* Executed in parallel */

      Sample array $\tilde{Q}_t^p(s, a) \sim \mathcal{N}(0, \frac{\beta_{N_{k-1}(\phi(s,a))}}{N_{k-1}(\phi(s,a))+1}), \forall (s, a)$

      /* Draw prior sample */

      $\mathcal{H}_k^p \leftarrow \{\}$

      **for** $(s, a, r, s') \in \mathcal{D}_k$ **do**

        /* Randomly perturb data */

        Sample $w^p(s, a) \sim \mathcal{N}(0, \frac{\beta_{N_{k-1}(\phi(s,a))}}{N_{k-1}(\phi(s,a))+1})$

        $\mathcal{H}_k^p \leftarrow \mathcal{H}_k^p \cup \{(s, a, r + w^p, s')\}$

      **end**

    **end**

    /* Estimate $Q$ on perturbed data */

    **for** $p = 1, \dots, N$ **do**

      /* Executed in Parallel */

      Define terminal value $\hat{Q}_{t_{k+1}}^p(\gamma) \leftarrow 0 \ \forall \gamma \in [\Gamma]$

      **for** $t = t_{k+1} - 1, \dots, t_k$ **do**

        /* Estimate Q on noisy data */

        $\hat{Q}_t^p \leftarrow \arg\min_{Q \in \mathbb{R}} \mathcal{L}(Q \mid \hat{Q}_{k-1}, \hat{Q}_{t+1}^p, \mathcal{H}_k^p, \xi, \eta, \alpha_{N_{k-1}(\gamma)}) + \|Q - \eta \alpha_{N_{k-1}(\gamma)} \tilde{Q}_t^p\|_2^2$

        $\hat{Q}_t^p \leftarrow \min\{\hat{Q}_t^p, \frac{1}{1-\eta}\}, \forall \gamma \in [\Gamma]$

      **end**

      $s_{t_k}^p = s_1^p, \forall p \in [N]$; $\mathcal{D}_k \leftarrow \emptyset$.

    **end**

    $\hat{Q}_k(\gamma) = \frac{\sum_{p=1}^N \sum_{t=t_k}^{t_{k+1} - 1} \mathbf{1}\{\phi(s_t^p, a_t^p) = \gamma\} \hat{Q}_{t_k}^p(\gamma)}{N_k(\gamma)}$,

    $\forall \gamma \in \Gamma$

    $t \leftarrow t_{k+1}, k \leftarrow k + 1$

  **end**

---

## C. Proofs for the Finite-Horizon Case

In this section, we prove the worst-case regret bound for the finite-horizon case. We use the following notations:

- $n_h^k(\gamma)$: the number of visits to aggregate state $\gamma$ at period $h$ from episodes $0$ to $k-1$.

- $N_{k,h}(\gamma)$: total number of agents who visit aggregate state $\gamma$ during episode $k$ and period $h$.

- $\pi^{kp}$: the greedy policy with respect to $\hat{Q}_{k,h}^p$, i.e. the policy that the agent $p$ follows to produce the trajectory $s_{k,1}^p, a_{k,1}^p, r_{k,1}^p, \ldots, s_{k,H}^p, a_{k,H}^p, r_{k,H}^p$.

- $\hat{V}_{k,h}^p(s)$: the state value function estimate at period $h$, induced by $\hat{Q}_{k,h}^p(\gamma)$ through

$$\hat{V}_{k,h}^p(s) = \max_{a \in \mathcal{A}} \hat{Q}_{k,h}^p(\phi_h(s,a)).$$

- $V^p(M, \pi)$: the value function corresponding to policy $\pi$ from the initial state $s_1^p$, where $s_1^p$ is the initial state of agent $p$ at the beginning of each episode. For the true MDP $M$ we have $V_1^\pi(s_1^p) := V^p(M, \pi)$.

- $\hat{V}_{k,1}^{\pi^{kp}}(s_1^p) := V^p(\overline{M}^{kp}, \pi^{kp})$: the value function corresponding to MDP $\overline{M}^{kp}$ with initial state $s_1^p$ and policy $\pi^{kp}$, where MDP $\overline{M}^{kp}$ is defined as (4).

For any $\gamma \in [\Gamma]$, we define the following events:

$$\mathcal{E}(\gamma) := \left\{ \left| \frac{1}{N_{k-1,h}(\gamma)} \sum_{j=1}^{N_{k-1,h}(\gamma)} \mathbf{1}\{\phi_h(s_{k-1,h}^j, a_{k-1,h}^j) = \gamma\}\{V_{h+1}^*(s_{k-1,h}^j) - P_h V_{h+1}^*(s_{k-1,h}^j)\} \right| \right.$$
$$\left. \le \frac{2H\sqrt{\log(2KHN/\delta)}}{\sqrt{N_{k-1,h}(\gamma)}} \right\}. \tag{18}$$

$$\mathcal{G}(\gamma) := \left\{ \left| \frac{1}{N_{k-1,h}(\gamma)} \sum_{j=1}^{N_{k-1,h}(\gamma)} \mathbf{1}\{\phi_h(s_{k-1,h}^j, a_{k-1,h}^j) = \gamma\} \tilde{Q}_{k-1,h}^j(s_{k-1,h}^j, a_{k-1,h}^j) \right| \right.$$
$$\left. \le 2 \frac{\sqrt{\beta_{k-1} \log(2KHN/\delta)}}{\sqrt{(N_{k-1,h}(\gamma) + 1)N_{k-1,h}(\gamma)}} \right\}. \tag{19}$$

### C.1. Proof of Finite-Horizon Main Result: Theorem 2

*Proof of Theorem 2.* Following the derivation of (37) and (38) in Lemma 11, we have

$$\sum_{\gamma \in [\Gamma]} \sum_{h=1}^H \sum_{k=1}^K \frac{1}{\sqrt{N_{k-1,h}(\gamma) + 1}} = \sum_{h=1}^H \sum_{k=1}^K \sum_{\gamma \in [\Gamma]} \sum_{j=1}^{N_{k-1,h}(\gamma)} \frac{1}{\sqrt{j}}$$
$$\le \sum_{k=1}^K \sum_{h=1}^H \sum_{\gamma \in [\Gamma]} 2\sqrt{N_{k,h}(\gamma)} \le 2\sqrt{KH\Gamma \sum_{k=1}^K \sum_{h=1}^H \sum_{\gamma \in [\Gamma]} N_{k,h}(\gamma)} = 2KH\sqrt{\Gamma N} \tag{20}$$

and

$$\begin{aligned}
\sum_{p=1}^N \sum_{k=1}^K \sum_{h=1}^H \frac{1}{\sqrt{N_{K-1,h}(\gamma_{kh}^p)}} &= \sum_{h=1}^H \sum_{k=1}^K \sum_{\gamma \in [\Gamma]} \sum_{j=1}^{N_{K-1,h}(\gamma)} \frac{1}{\sqrt{j}} \\
&\le \sum_{h=1}^H \sum_{k=1}^K \sum_{\gamma \in [\Gamma]} 2\sqrt{N_{k-1,h}(\gamma)} \\
&\le 2\sqrt{HK\Gamma \sum_{h=1}^H \sum_{k=1}^K \sum_{\gamma \in [\Gamma]} N_{k-1,h}(\gamma)} \\
&= 2HK\sqrt{\Gamma N}.
\end{aligned}$$

Under events $\mathcal{E}(\gamma_{kh}^p), \mathcal{G}(\gamma_{kh}^p), \forall \gamma_{kh}^p \in [\Gamma]$, we have

$$\text{Regret}(M, K, N, \pi, \text{RLSVI}_{\beta,\alpha,\xi}) = \sum_{k=1}^{K} \sum_{p=1}^{N} V_1^*(s_1^p) - V_{\pi^{kp}}^{\eta}(s_1^p) \leq \sum_{k=1}^{K} \sum_{p=1}^{N} \hat{V}_{k,1}(s_1^p) - V_{\pi^{kp}}^{\eta}(s_1^p).$$

Recall from Lemma 11 that when $\mathcal{E}(\gamma_{kh}^p)$ and $\mathcal{G}(\gamma_{kh}^p)$ hold for all $\gamma_{kh}^p \in [\Gamma]$, with probability $1 - 2\delta$, we have

$$\begin{aligned}
&\sum_{p=1}^{N} \sum_{k=1}^{K} \hat{V}_{k,1}^p(s_1^p) - V_{\pi^{kp}}^{\eta}(s_1^p) \\
&\leq 2\epsilon KHN + 8KH^2\sqrt{\Gamma N}\sqrt{\log(2KHN/\delta)} + 32H^2\sqrt{K\Gamma N}\sqrt{\log(HKN/\delta)} \\
&\quad + 2KH^{5/2}\Gamma\sqrt{N}\sqrt{\log(2KH\Gamma)}\sqrt{\log(2KHN/\delta)}.
\end{aligned}$$

Note that by (18), (19) and the first statement of Lemma 10, we have

$$\begin{aligned}
&\mathbb{P}(\mathcal{E}(\gamma_{kh}^p), \mathcal{G}(\gamma_{kh}^p), \gamma_{kh}^p, \forall k \in [K], h \in [H], p \in [N]) \\
&\geq 1 - \sum_{k \in [K], h \in [H], p \in [N]} (\mathbb{P}(\mathcal{E}(\gamma_{kh}^p)^c) + \mathbb{P}(\mathcal{G}(\gamma_{kh}^p)^c)) \\
&\geq 1 - 2NKH\delta/(2KHN) \geq 1 - \delta.
\end{aligned} \tag{21}$$

Hence with probability $1 - 3\delta$, we have

$$\begin{aligned}
&\text{Regret}(M, K, N, \pi, \text{RLSVI}_{\beta,\alpha,\xi}) \\
&\leq 2\epsilon KHN + 8KH^2\sqrt{\Gamma N}\sqrt{\log(2KHN/\delta)} + 32H^2\sqrt{K\Gamma N}\sqrt{\log(HKN/\delta)} \\
&\quad + 2KH^{5/2}\Gamma\sqrt{N}\sqrt{\log(2KH\Gamma)}\sqrt{\log(2KHN/\delta)}.
\end{aligned}$$

So we obtain (9).

$\square$

## C.2. Lemmas for State-Aggregation Results

**Lemma 8** (Concentration Bound). *Conditioning on $\mathcal{D}_k = \cup_{p \in [N]} \sum_{h \in [H]} \{s_{k-1,h}^p, a_{k-1,h}^p\}$ as the trajectories of episode $k - 1$ across all agents, for every possible $\gamma_{kh}^p = \phi_h(s_{k,h}^p, a_{k,h}^p)$ in episode $k$, event $\mathcal{E}(\gamma_{kh}^p)$ defined in (18) and event $\mathcal{G}(\gamma_{kh}^p)$ both hold with probability $1 - \delta/(2KHN)$.*

*Proof of Lemma 8.* By Höeffding's inequality, conditional on the trajectory during episode $k - 1$, with probability $1 - \delta/(2KHN)$, we have

$$\left| \frac{1}{N_{k-1,h}(\gamma_{kh}^p)} \sum_{j=1}^{N_{k-1,h}(\gamma_{kh}^p)} \{V_{h+1}^*(s_{k-1,h+1}^j) - P_h V_{h+1}^*(s_{k-1,h}^j)\} \right| \leq \frac{2H\sqrt{\log(2KHN/\delta)}}{\sqrt{N_{k-1,h}(\gamma_{kh}^p)}}. \tag{22}$$

Additionally, recall from Algorithm 3 that for $k \geq 2$, the random perturbation during episode $k - 1$ is drawn from normal distribution $\tilde{Q}_{k-1,h}^j(s_{k-1,h}^j, a_{k-1,h}^j) \sim \mathcal{N}(0, \frac{\beta_{k-1}}{N_{k-2,h}(\phi_h(s_{k-1,h}^j, a_{k-1,h}^j))+1})$, and since $\phi_h(s_{k-1,h}^j, a_{k-1,h}^j) = \gamma_{kh}^p$ in (24), we have $\tilde{Q}_{k-1,h}^j(s_{k-1,h}^j, a_{k-1,h}^j) \sim \mathcal{N}(0, \frac{\beta_{k-1}}{N_{k-2,h}(\gamma_{kh}^p)+1})$. Also note that the random perturbations $\tilde{Q}_{k-1,h}^j(s_{k-1,h}^j, a_{k-1,h}^j)$ are i.i.d. across $j \in [N]$, hence by Höeffding bound, conditioning on $\mathcal{D}_k$, with probability $1 - \delta/(2KHN)$, we have

$$\left| \frac{1}{N_{k-1,h}(\gamma_{kh}^p)} \sum_{j=1}^{N_{k-1,h}(\gamma_{kh}^p)} \tilde{Q}_{k-1,h}^j(s_{k-1,h}^j, a_{k-1,h}^j) \right| \leq 2\sqrt{\frac{\beta_{k-1}}{N_{k-2,h}(\gamma_{kh}^p)+1}}\sqrt{\frac{\log(2KHN/\delta)}{N_{k-1,h}(\gamma_{kh}^p)}}.$$

Thus the result follows.

$\square$

**Lemma 9** (Optimism). *When events $\mathcal{E}(\gamma_{kh}^p)$ and $\mathcal{G}(\gamma_{kh}^p)$ hold for all $\gamma_{kh}^p$ in episode $k$ and for all $k \in [K], p \in [N]$, the on-policy error $\hat{V}_{k,h}^p(s) - V_h^*(s)$ is lower bounded by zero for any $s \in \mathcal{S}, k \in [K], p \in [N]$.*

*Proof of Lemma 9.* Recall from Algorithm 3 that the unclipped value function estimates $\bar{Q}^p_{k,h}(\cdot)$ in episode $k$ can be written as

$$\bar{Q}^p_{k,h}(\gamma) = \arg\min_{Q\in\mathbb{R}} \sum_{(s,a):\phi_h(s,a)=\gamma}(Q - \xi_{N_{k-1,h}(\gamma)} - (1 - \alpha_{N_{k-1,h}(\gamma)})\hat{Q}_{k-1,h}(\gamma)$$
$$-\alpha_{N_{k-1,h}(\gamma)}\{r(s,a) + \max_{a'\in\mathcal{A}}\hat{Q}^p_{k,h+1}(s',a')\})^2 + \left\|Q - \alpha_{N_{k-1,h}(\gamma)}\tilde{Q}^p_{k,h}\right\|^2_2.$$

By first-order condition, we have

$$0 = 2\sum_{(s,a):\phi_h(s,a)=\gamma}(\bar{Q}^p_{k,h}(\gamma) - \xi_{N_{k-1,h}(\gamma)} - (1 - \alpha_{N_{k-1,h}(\gamma)})\hat{Q}_{k-1,h}(\gamma)$$
$$-\alpha_{N_{k-1,h}(\gamma)}\{r(s,a) + \max_{a'\in\mathcal{A}}\hat{Q}^p_{k,h+1}(s',a')\})$$
$$+2\sum_{(s,a)\in\mathcal{D}^k_h:\phi_h(s,a)=\gamma}(\bar{Q}^p_{k,h}(\gamma) - \alpha_{N_{k-1,h}(\gamma)}\tilde{Q}^p_{k,h}(s,a)),$$

so by calculation we have

$$\bar{Q}^p_{k,h}(\gamma)$$
$$= \xi_{N_{k-1,h}(\gamma)} + \frac{1}{N_{k-1,h}(\gamma)}\sum_{(s,a)\in\mathcal{D}^k_h:\phi_h(s,a)=\gamma}(1 - \alpha_{N_{k-1,h}(\gamma)})\hat{Q}_{k-1,h}(\gamma)$$
$$+\alpha_{N_{k-1,h}(\gamma)}(r(s,a) + \max_{a'\in\mathcal{A}}\hat{Q}^p_{k,h+1}(s',a')) + \alpha_{N_{k-1,h}(\gamma)}\tilde{Q}^p_{k,h}(s,a))$$
$$= \xi_{N_{k-1,h}(\gamma)} + \frac{1}{N_{k-1,h}(\gamma)}\sum_{p=1}^{N}\sum_{(s^p_{k-1,h},a^p_{k-1,h})\in\mathcal{D}^p_{kh}:\phi_h(s,a)=\gamma}(1 - \alpha_{N_{k-1,h}(\gamma)})\hat{Q}_{k-1,h}(\gamma)$$
$$+\alpha_{N_{k-1,h}(\gamma)}(r(s^p_{k-1,h},a^p_{k-1,h}) + \hat{V}^p_{k,h+1}(s^p_{k-1,h+1})) + \alpha_{N_{k-1,h}(\gamma)}\tilde{Q}^p_{k,h}(s^p_{k-1,h},a^p_{k-1,h})$$
$$= \xi_{N_{k-1,h}(\gamma)} + (1 - \alpha_{N_{k-1,h}(\gamma)})\hat{Q}_{k-1,h}(\gamma)$$
$$+\frac{\alpha_{N_{k-1,h}(\gamma)}}{N_{k-1,h}(\gamma)}\sum_{p=1}^{N}\sum_{(s^p_{k-1,h},a^p_{k-1,h})\in\mathcal{D}^p_{k,h}:\phi_h(s,a)=\gamma}(r(s^p_{k-1,h},a^p_{k-1,h}) + \hat{V}^p_{k,h+1}(s^p_{k-1,h+1}) + \tilde{Q}^p_{k,h}(s^p_{k-1,h},a^p_{k-1,h}))$$
$$= \xi_{N_{k-1,h}(\gamma)} + (1 - \alpha_{N_{k-1,h}(\gamma)})\hat{Q}_{k-1,h}(\gamma)$$
$$+\frac{\alpha_{N_{k-1,h}(\gamma)}}{N_{k-1,h}(\gamma)}\sum_{p=1}^{N_{k-1,h}(\gamma)}(r(s^p_{k-1,h},a^p_{k-1,h}) + \hat{V}^p_{k,h+1}(s^p_{k-1,h+1}) + \tilde{Q}^p_{k,h}(s^p_{k-1,h},a^p_{k-1,h}))$$
$$= \xi_{N_{k-1,h}(\gamma)} + \frac{1}{N_{k-1,h}(\gamma)}\sum_{p=1}^{N_{k-1,h}(\gamma)}(1 - \alpha_{N_{k-1,h}(\gamma)})\hat{Q}^p_{k-1,h}(\gamma)$$
$$+\alpha_{N_{k-1,h}(\gamma)}(r(s^p_{k-1,h},a^p_{k-1,h}) + \hat{V}^p_{k,h+1}(s^p_{k-1,h+1}) + \tilde{Q}^p_{k,h}(s^p_{k-1,h},a^p_{k-1,h})),$$

where in the above derivation we used the fact that $\sum_{p=1}^{N}\mathbf{1}\{\phi_h(s^p_{k-1,h},a^p_{k-1,h}) = \gamma\} = N_{k-1,h}(\gamma)$.

Thus for $\gamma^p_{kh} = \phi_h(s^p_{k,h},a^p_{k,h})$, with $\phi_h(s^j_{k-1,h},a^j_{k-1,h}) = \gamma^p_{kh}$ in the following, we have

$$\bar{Q}_{k,h}^p(s_{k,h}^p, a_{k,h}^p) - Q_h^*(s_{k,h}^p, a_{k,h}^p)$$

$$= \xi_{N_{k-1,h}(\gamma)} + \frac{1}{N_{k-1,h}(\gamma_{kh}^p)} \sum_{j=1}^{N_{k-1,h}(\gamma_{kh}^p)} (1 - \alpha_{N_{k-1,h}(\gamma_{kh}^p)})(\hat{Q}_{k-1,h}^j(s_{k-1,h}^j, a_{k-1,h}^j) - Q_h^*(s_{k,h}^p, a_{k,h}^p))$$

$$+ \alpha_{N_{k-1,h}(\gamma_{kh}^p)}[\{r_{k-1,h}^j(s_{k-1,h}^j, a_{k-1,h}^j) + \hat{V}_{k-1,h+1}^j(s_{k-1,h+1}^j) + \tilde{Q}_{k-1,h}^j(s_{k-1,h}^j, a_{k-1,h}^j)\}$$

$$- Q_h^*(s_{k,h}^p, a_{k,h}^p)]$$

$$= \xi_{N_{k-1,h}(\gamma)} + \frac{1}{N_{k-1,h}(\gamma_{kh}^p)} \sum_{j=1}^{N_{k-1,h}(\gamma_{kh}^p)} (1 - \alpha_{N_{k-1,h}(\gamma_{kh}^p)})(\hat{Q}_{k-1,h}^j(s_{k-1,h}^j, a_{k-1,h}^j) - Q_h^*(s_{k,h}^p, a_{k,h}^p))$$

$$+ \frac{\alpha_{N_{k-1,h}(\gamma_{kh}^p)}}{N_{k-1,h}(\gamma_{kh}^p)} \sum_{p=1}^{N_{k-1,h}(\gamma_{kh}^p)} \{[r_{k-1,h}^j(s_{k-1,h}^j, a_{k-1,h}^j) + \hat{V}_{k-1,h+1}^j(s_{k-1,h+1}^j) + \tilde{Q}_{k-1,h}^j(s_{k-1,h}^j, a_{k-1,h}^j)]$$

$$- Q_h^*(s_{k-1,h}^j, a_{k-1,h}^j)\}$$

$$+ \frac{\alpha_{N_{k-1,h}(\gamma_{kh}^p)}}{N_{k-1,h}(\gamma_{kh}^p)} \sum_{j=1}^{N_{k-1,h}(\gamma_{kh}^p)} \underbrace{\{Q_h^*(s_{k-1,h}^j, a_{k-1,h}^j) - Q_h^*(s_{k,h}^p, a_{k,h}^p)\}}_{\geq -\epsilon}$$

$$\geq -\epsilon + \xi_{N_{k-1,h}(\gamma)} + \frac{1}{N_{k-1,h}(\gamma_{kh}^p)} \sum_{j=1}^{N_{k-1,h}(\gamma_{kh}^p)} (1 - \alpha_{N_{k-1,h}(\gamma_{kh}^p)})(\hat{Q}_{k-1,h}^j(s_{k-1,h}^j, a_{k-1,h}^j - Q_h^*(s_{k-1,h}^j, a_{k-1,h}^j))$$

$$+ \frac{\alpha_{N_{k-1,h}(\gamma_{kh}^p)}}{N_{k-1,h}(\gamma_{kh}^p)} \sum_{j=1}^{N_{k-1,h}(\gamma_{kh}^p)} \{[r_{k-1,h}^j(s_{k-1,h}^j, a_{k-1,h}^j) + \hat{V}_{k-1,h+1}^j(s_{k-1,h+1}^j) + \tilde{Q}_{k-1,h}^j(s_{k-1,h}^j, a_{k-1,h}^j)]$$

$$- Q_h^*(s_{k-1,h}^j, a_{k-1,h}^j)\}$$

$$\tag{23}$$

where we use the definition of $\epsilon$-error aggregation as in Definition 1 in the last inequality above, and the right hand side of (23) is equal to

$$\text{RHS of (23)} =_{(1)} -\epsilon + \xi_{N_{k-1,h}(\gamma)} + \frac{(1 - \alpha_{N_{k-1,h}(\gamma_{kh}^p)})}{N_{k-1,h}(\gamma)} \sum_{j=1}^{N_{k-1,h}(\gamma_{kh}^p)} (\hat{Q}_{k-1,h}^j(s_{k-1,h}^j, a_{k-1,h}^j) - Q_h^*(s_{k-1,h}^j, a_{k-1,h}^j))$$

$$+ \frac{\alpha_{N_{k-1,h}(\gamma_{kh}^p)}}{N_{k-1,h}(\gamma_{kh}^p)} \sum_{j=1}^{N_{k-1,h}(\gamma_{kh}^p)} \tilde{Q}_{k-1,h}^j(s_{k-1,h}^j, a_{k-1,h}^j)$$

$$+ \frac{\alpha_{N_{k-1,h}(\gamma_{kh}^p)}}{N_{k-1,h}(\gamma_{kh}^p)} \sum_{j=1}^{N_{k-1,h}(\gamma_{kh}^p)} \{\hat{V}_{k-1,h+1}^j(s_{k-1,h+1}^j) - V_{h+1}^*(s_{k-1,h+1}^j)\}$$

$$+ \frac{\alpha_{N_{k-1,h}(\gamma_{kh}^p)}}{N_{k-1,h}(\gamma_{kh}^p)} \sum_{j=1}^{N_{k-1,h}(\gamma_{kh}^p)} \{V_{h+1}^*(s_{k-1,h+1}^j) - P_h V_{h+1}^*(s_{k-1,h}^j)\}$$

$$=_{(2)} -\epsilon + \xi_{N_{k-1,h}(\gamma)} + \frac{\alpha_{N_{k-1,h}(\gamma_{kh}^p)}}{N_{k-1,h}(\gamma_{kh}^p)} \sum_{j=1}^{N_{k-1,h}(\gamma_{kh}^p)} \tilde{Q}_{k-1,h}^j(s_{k-1,h}^j, a_{k-1,h}^j)$$

$$+ \frac{1}{N_{k-1,h}(\gamma_{kh}^p)} \sum_{j=1}^{N_{k-1,h}(\gamma_{kh}^p)} \{\hat{V}_{k-1,h+1}^j(s_{k-1,h+1}^j) - V_{h+1}^*(s_{k-1,h+1}^j)\}$$

$$+ \frac{\alpha_{N_{k-1,h}(\gamma_{kh}^p)}}{N_{k-1,h}(\gamma_{kh}^p)} \sum_{j=1}^{N_{k-1,h}(\gamma_{kh}^p)} \{V_{h+1}^*(s_{k-1,h+1}^j) - P_h V_{h+1}^*(s_{k-1,h}^j)\}$$

$$\tag{24}$$

where the equalities (1) and (2) use the fact that

$$Q_h(s', a') = r_h(s', a') + P_h V_{h+1}^*(s', a'), \quad \forall (s', a') \in \mathcal{S} \times \mathcal{A}.$$

Suppose events $\mathcal{E}(\gamma_{kh}^p)$ and $\mathcal{G}(\gamma_{kh}^p)$ hold for all $\gamma_{kh}^p$ in episode $k$, then by (18),(19), we have

$$\frac{\alpha_{N_{k-1,h}(\gamma_{kh}^p)}}{N_{k-1,h}(\gamma_{kh}^p)} \sum_{j=1}^{N_{k-1,h}(\gamma_{kh}^p)} \{V_{h+1}^*(s_{k-1,h+1}^j) - P_h V_{h+1}^*(s_{k-1,h}^j)\} \geq -\frac{2\alpha_{N_{k-1,h}(\gamma_{kh}^p)} H \sqrt{\log(2KHN/\delta)}}{\sqrt{\max\{N_{k-1,h}(\gamma_{kh}^p), 1\}}},$$

and

$$\frac{\alpha_{N_{k-1,h}(\gamma_{kh}^p)}}{N_{k-1,h}(\gamma_{kh}^p)} \sum_{j=1}^{N_{k-1,h}(\gamma_{kh}^p)} \tilde{Q}_{k-1,h}^j(s_{k-1,h}^j, a_{k-1,h}^j) \geq -\frac{2\alpha_{N_{k-1,h}(\gamma_{kh}^p)} \sqrt{\beta_{k-1} \log(2KHN/\delta)}}{\sqrt{(N_{k-1,h}(\gamma_{kh}^p) + 1) \max\{N_{k-1,h}(\gamma_{kh}^p), 1\}}}.$$

So recall from (7)

$$\xi_{N_{k-1,h}(\gamma_{kh}^p)} = \epsilon + \frac{2\alpha_{N_{k-1,h}(\gamma_{kh}^p)} H \sqrt{\log(2KHN/\delta)}}{\sqrt{\max\{N_{k-1,h}(\gamma_{kh}^p), 1\}}} + \frac{2\alpha_{N_{k-1,h}(\gamma_{kh}^p)} \sqrt{\beta_{k-1} \log(2KHN/\delta)}}{\sqrt{(N_{k-1,h}(\gamma_{kh}^p) + 1) \max\{N_{k-1,h}(\gamma_{kh}^p), 1\}}},$$

thus we have

$$\bar{Q}_{k,h}^p(s_{k,h}^p, a_{k,h}^p) - Q_h^*(s_{k,h}^p, a_{k,h}^p) \geq \frac{1}{N_{k-1,h}(\gamma_{kh}^p)} \sum_{j=1}^{N_{k-1,h}(\gamma_{kh}^p)} \{\hat{V}_{k-1,h+1}^j(s_{k-1,h+1}^j) - V_{h+1}^*(s_{k-1,h+1}^j)\}. \tag{25}$$

Note that when $N_{k-1,h}(\gamma) = 0$, then $\bar{Q}_{k,h}^p(\gamma) = \hat{Q}_{k,h}^p(\gamma) = H$, and we define terminal values as $H$ with $\bar{Q}_{k,h+1}^p(\gamma) = \hat{Q}_{k,h+1}^p(\gamma) = H$, so by plugging in $h = H$ in (25), for any possible $\gamma \in [\Gamma]$, we have $\bar{Q}_{k,H}^p(\gamma) \geq Q_H^*(\gamma)$. Thus

$$\hat{V}_{k,H}^p(s) \geq V_H^*(s), \ \forall s \in \mathcal{S}, k \in [K], p \in [N].$$

Furthermore, suppose that at period $h$, we have

$$\hat{V}_{k,h}^p(s) \geq V_h^*(s), \ \forall s \in \mathcal{S}, k \in [K], p \in [N],$$

then from (25) we have

$$\bar{Q}_{k,h-1}^p(s, a) - Q_h^*(s, a) \geq \frac{1}{N_{k-1,h}(\phi_h(s, a)} \sum_{j=1}^{N_{k-1,h}(\phi_h(s,a))} \{\hat{V}_{k-1,h+1}^j(s_{k-1,h+1}^j) - V_{h+1}^*(s_{k-1,h+1}^j)\} \geq 0,$$

which implies that

$$\bar{Q}_{k,h-1}^p(s, a) \geq Q_h^*(s, a), \ \forall (s, a) \in \mathcal{S} \times \mathcal{A}, k \in [K], p \in [N].$$

By maximizing over $a \in \mathcal{A}$ on both sides above, we have

$$\hat{V}_{k-1,h-1}^p \geq V_{h-1}^*(s), \ \forall s \in \mathcal{S}, k \in [K].$$

Thus by induction, we conclude that when events $\mathcal{E}(\gamma_{kh}^p)$ and $\mathcal{G}(\gamma_{kh}^p)$ hold for all $\gamma_{kh}^p$ in episode $k$ for all $k \in [K]$, we have

$$\hat{V}_{k,h}^p(s) \geq V_h^*(s), \ \forall s \in \mathcal{S}, k \in [K], p \in [N].$$

$\square$

**Lemma 10.** *Suppose events $\mathcal{E}(\gamma_{kh}^p)$ and $\mathcal{G}(\gamma_{kh}^p)$ hold for all $\gamma_{kh}^p$, for all $k \in [K]$ and $p \in [N]$, then we have*

$$\sum_{p=1}^N \sum_{k=1}^K \left[ \hat{V}_{k,h}^p(s_{k,h}^p) - V_h^*(s_{k,h}^p) \right] \leq 2\epsilon K N(H - h + 1)$$

$$+ 4H\sqrt{\log(2KHN/\delta)} \sum_{k=1}^K \sum_{p=1}^N \sum_{\ell=h}^H \frac{1}{\sqrt{N_{k-1,\ell}(\gamma_{k\ell}^p)}} \frac{1}{1 + N_{k-1,\ell}(\gamma_{k\ell}^p)}$$

$$+ 4 \sum_{\ell=h}^H \sum_{p=1}^N \sum_{k=2}^K \frac{\sqrt{\beta_{k-1} \log(2KHN/\delta)}}{\sqrt{(n_{k-1}^h(\gamma_{kh}^p) + 1) N_{k-2,h}(\gamma_{kh}^p)}} \frac{1}{1 + N_{k-1,\ell}(\gamma_{k\ell}^p)}.$$

*Proof of Lemma 10.* By (23) we have

$$\bar{Q}^p_{k,h}(s^p_{k,h}, a^p_{k,h}) - Q^*_h(s^p_{k,h}, a^p_{k,h})$$

$$= \xi_{N_{k-1,h}(\gamma)} + \frac{1}{N_{k-1,h}(\gamma^p_{kh})} \sum_{j=1}^{N_{k-1,h}(\gamma^p_{kh})} (1 - \alpha_{N_{k-1,h}(\gamma^p_{kh})})(\hat{Q}^j_{k-1,h}(s^j_{k-1,h}, a^j_{k-1,h}) - Q^*_h(s^p_{k,h}, a^p_{k,h}))$$

$$+ \alpha_{N_{k-1,h}(\gamma^p_{kh})}[\{r^j_{k-1,h}(s^j_{k-1,h}, a^j_{k-1,h}) + \hat{V}^j_{k-1,h+1}(s^j_{k-1,h+1}) + \tilde{Q}^j_{k-1,h}(s^j_{k-1,h}, a^j_{k-1,h})\}$$
$$- Q^*_h(s^p_{k,h}, a^p_{k,h})]$$

$$= \xi_{N_{k-1,h}(\gamma)} + \frac{1}{N_{k-1,h}(\gamma^p_{kh})} \sum_{j=1}^{N_{k-1,h}(\gamma^p_{kh})} (1 - \alpha_{N_{k-1,h}(\gamma^p_{kh})})(\hat{Q}^j_{k-1,h}(s^j_{k-1,h}, a^j_{k-1,h}) - Q^*_h(s^p_{k,h}, a^p_{k,h}))$$

$$+ \frac{\alpha_{N_{k-1,h}(\gamma^p_{kh})}}{N_{k-1,h}(\gamma^p_{kh})} \sum_{p=1}^{N_{k-1,h}(\gamma^p_{kh})} \{[r^j_{k-1,h}(s^j_{k-1,h}, a^j_{k-1,h}) + \hat{V}^j_{k-1,h+1}(s^j_{k-1,h+1}) + \tilde{Q}^j_{k-1,h}(s^j_{k-1,h}, a^j_{k-1,h})]$$
$$- Q^*_h(s^j_{k-1,h}, a^j_{k-1,h})\}$$

$$+ \frac{\alpha_{N_{k-1,h}(\gamma^p_{kh})}}{N_{k-1,h}(\gamma^p_{kh})} \sum_{j=1}^{N_{k-1,h}(\gamma^p_{kh})} \underbrace{\{Q^*_h(s^j_{k-1,h}, a^j_{k-1,h}) - Q^*_h(s^p_{k,h}, a^p_{k,h})\}}_{\leq \epsilon}$$

$$\leq \epsilon + \xi_{N_{k-1,h}(\gamma)} + \frac{1}{N_{k-1,h}(\gamma^p_{kh})} \sum_{j=1}^{N_{k-1,h}(\gamma^p_{kh})} (1 - \alpha_{N_{k-1,h}(\gamma^p_{kh})})(\hat{Q}^j_{k-1,h}(s^j_{k-1,h}, a^j_{k-1,h} - Q^*_h(s^j_{k-1,h}, a^j_{k-1,h}))$$

$$+ \frac{\alpha_{N_{k-1,h}(\gamma^p_{kh})}}{N_{k-1,h}(\gamma^p_{kh})} \sum_{j=1}^{N_{k-1,h}(\gamma^p_{kh})} \{[r^j_{k-1,h}(s^j_{k-1,h}, a^j_{k-1,h}) + \hat{V}^j_{k-1,h+1}(s^j_{k-1,h+1}) + \tilde{Q}^j_{k-1,h}(s^j_{k-1,h}, a^j_{k-1,h})]$$
$$- Q^*_h(s^j_{k-1,h}, a^j_{k-1,h})\}$$

$$\tag{26}$$

where we use the definition of $\epsilon$-error aggregation as in Definition 1 in the last inequality above, and the right hand side of (26) is equal to

$$\text{RHS of (26)} \quad =_{(1)} \epsilon + \xi_{N_{k-1,h}(\gamma)} + \frac{(1 - \alpha_{N_{k-1,h}(\gamma^p_{kh})})}{N_{k-1,h}(\gamma)} \sum_{j=1}^{N_{k-1,h}(\gamma^p_{kh})} (\hat{Q}^j_{k-1,h}(s^j_{k-1,h}, a^j_{k-1,h}) - Q^*_h(s^j_{k-1,h}, a^j_{k-1,h}))$$

$$+ \frac{\alpha_{N_{k-1,h}(\gamma^p_{kh})}}{N_{k-1,h}(\gamma^p_{kh})} \sum_{j=1}^{N_{k-1,h}(\gamma^p_{kh})} \tilde{Q}^j_{k-1,h}(s^j_{k-1,h}, a^j_{k-1,h})$$

$$+ \frac{\alpha_{N_{k-1,h}(\gamma^p_{kh})}}{N_{k-1,h}(\gamma^p_{kh})} \sum_{j=1}^{N_{k-1,h}(\gamma^p_{kh})} \{\hat{V}^j_{k-1,h+1}(s^j_{k-1,h+1}) - V^*_{h+1}(s^j_{k-1,h+1})\}$$

$$+ \frac{\alpha_{N_{k-1,h}(\gamma^p_{kh})}}{N_{k-1,h}(\gamma^p_{kh})} \sum_{j=1}^{N_{k-1,h}(\gamma^p_{kh})} \{V^*_{h+1}(s^j_{k-1,h+1}) - P_h V^*_{h+1}(s^j_{k-1,h})\}$$

$$=_{(2)} \epsilon + \xi_{N_{k-1,h}(\gamma)} + \frac{\alpha_{N_{k-1,h}(\gamma^p_{kh})}}{N_{k-1,h}(\gamma^p_{kh})} \sum_{j=1}^{N_{k-1,h}(\gamma^p_{kh})} \tilde{Q}^j_{k-1,h}(s^j_{k-1,h}, a^j_{k-1,h})$$

$$+ \frac{1}{N_{k-1,h}(\gamma^p_{kh})} \sum_{j=1}^{N_{k-1,h}(\gamma^p_{kh})} \{\hat{V}^j_{k-1,h+1}(s^j_{k-1,h+1}) - V^*_{h+1}(s^j_{k-1,h+1})\}$$

$$+ \frac{\alpha_{N_{k-1,h}(\gamma^p_{kh})}}{N_{k-1,h}(\gamma^p_{kh})} \sum_{j=1}^{N_{k-1,h}(\gamma^p_{kh})} \{V^*_{h+1}(s^j_{k-1,h+1}) - P_h V^*_{h+1}(s^j_{k-1,h})\}$$

$$\tag{27}$$

where the equalities (1) and (2) above use the fact that

$$Q_h(s', a') = r_h(s', a') + P_h V^*_{h+1}(s', a'), \quad \forall (s', a') \in \mathcal{S} \times \mathcal{A}.$$

Now suppose $\mathcal{E}(\gamma_{kh}^p)$ and $\mathcal{G}(\gamma_{kh}^p)$ holds for all aggregated states $\gamma_{kh}^p$ during period $k$ for all $p \in [N]$.

By definition, note that

$$\hat{V}_{k,h}^p(s_{k,h}^p) - V_h^*(s_{k,h}^p) \le \hat{Q}_{k,h}^p(s_{k,h}^p, a_{k,h}^p) - Q_h^*(s_{k,h}^p, a_{k,h}^p) \le \bar{Q}_{k,h}^p(s_{k,h}^p, a_{k,h}^p) - Q_h^*(s_{k,h}^p, a_{k,h}^p), \tag{28}$$

Denote

$$\Delta_{k,h}^p = \hat{V}_{k,h}^p(s_{k,h}^p) - V_h^*(s_{k,h}^p), \tag{29}$$

then under events $\mathcal{E}(\gamma_{kh}^p)$ and $\mathcal{G}(\gamma_{kh}^p)$,

$$
\begin{aligned}
\bar{Q}_{k,h}^p(s_{k,h}^p, a_{k,h}^p) - Q_h^*(s_{k,h}^p, a_{k,h}^p) \quad &\le \epsilon + \xi_{N_{k-1,h}(\gamma)} + \frac{2\alpha_{N_{k-1,h}(\gamma_{kh}^p)}\sqrt{\beta_{k-1}\log(2KHN/\delta)}}{\sqrt{(N_{k-2,h}(\gamma_{kh}^p)+1)N_{k-1,h}(\gamma_{kh}^p)}} \\
&\quad + \frac{\alpha_{N_{k-1,h}(\gamma_{kh}^p)}}{N_{k-1,h}(\gamma_{kh}^p)} \sum_{j=1}^{N_{k-1,h}(\gamma_{kh}^p)} \Delta_{k,h+1}^j + \frac{2\alpha_{N_{k-1,h}(\gamma_{kh}^p)}H\sqrt{\log(2KHN/\delta)}}{\sqrt{N_{k-1,h}(\gamma_{kh}^p)}}.
\end{aligned}
\tag{30}
$$

Since $\mathcal{E}(\gamma_{kh}^p)$ and $\mathcal{G}(\gamma_{kh}^p)$ hold for all $\gamma_{kh}^p \in [\Gamma]$, and recall from (7)

$$\xi_{N_{k-1,h}(\gamma_{kh}^p)} = \epsilon + \frac{2\alpha_{N_{k-1,h}(\gamma_{kh}^p)}H\sqrt{\log(2KHN/\delta)}}{\sqrt{\max\{N_{k-1,h}(\gamma_{kh}^p), 1\}}} + \frac{2\alpha_{N_{k-1,h}(\gamma_{kh}^p)}\sqrt{\beta_{k-1}\log(2KHN/\delta)}}{\sqrt{(N_{k-1,h}(\gamma_{kh}^p)+1)\max\{N_{k-1,h}(\gamma_{kh}^p), 1\}}},$$

so we have

$$
\begin{aligned}
\sum_{p=1}^N \sum_{k=1}^K \Delta_{k,h}^p(s_{k,h}^p) \quad &\le \sum_{p=1}^N \sum_{k=1}^K Q_h^*(s_{k,h}^p, a_{k,h}^p) - \bar{Q}_{k,h}^p(s_{k,h}^p, a_{k,h}^p) \\
&\le 2KN\epsilon + 4\sum_{p=1}^N \sum_{k=1}^K \frac{\alpha_{N_{k-1,h}(\gamma_{kh}^p)}H\sqrt{\log(2KHN/\delta)}}{\sqrt{N_{k-1,h}(\gamma_{kh}^p)}} \\
&\quad + 4\sum_{p=1}^N \sum_{k=2}^K \frac{\alpha_{N_{k-1,h}(\gamma_{kh}^p)}\sqrt{\beta_{k-1}\log(2KHN/\delta)}}{\sqrt{(N_{k-2,h}(\gamma_{kh}^p)+1)N_{k-1,h}(\gamma_{kh}^p)}} \\
&\quad + \sum_{p=1}^N \sum_{k=1}^K \frac{\alpha_{N_{k-1,h}(\gamma_{kh}^p)}}{N_{k-1,h}(\gamma_{kh}^p)} \sum_{j=1}^{N_{k-1,h}(\gamma_{kh}^p)} \Delta_{k,h+1}^j.
\end{aligned}
\tag{31}
$$

For any $\gamma \in [\Gamma]$, for $s_{k,h}^p, a_{k,h}^p$ such that $\phi_h(s_{k,h}^p, a_{k,h}^p) = \gamma$, denote

$$\bar{\Delta}_{k,h+1}(\gamma) = \frac{1}{N_{k-1,h}(\gamma)} \sum_{p=1}^{N_{k-1,h}(\gamma)} \Delta_{k,h+1}^p.$$

Note that

$$
\begin{aligned}
&\sum_{p=1}^N \sum_{k=1}^K \frac{\alpha_{N_{k-1,h}(\gamma_{kh}^p)}}{N_{k-1,h}(\gamma_{kh}^p)} \sum_{j=1}^{N_{k-1,h}(\gamma_{kh}^p)} \Delta_{k,h+1}^j \\
&= \sum_{k=1}^K N_{k-1,h}(\gamma_{kh}^p)\alpha_{N_{k-1,h}(\gamma_{kh}^p)}\bar{\Delta}_{k,h+1} = \sum_{k=1}^K \alpha_{N_{k-1,h}(\gamma_{kh}^p)} \sum_{p=1}^{N_{k-1,h}(\gamma_{kh}^p)} \Delta_{k,h+1}^p \\
&\le \sum_{k=1}^K \sum_{p=1}^N \Delta_{k,h+1}^p,
\end{aligned}
\tag{32}
$$

where the last inequality above follows from Lemma 9.

Hence by above recursion and (31) we have that when $\mathcal{E}(\gamma_{kh}^p)$ and $\mathcal{G}(\gamma_{kh}^p)$ hold for all $\gamma_{kh}^p \in [\Gamma]$, we have

$$\sum_{p=1}^{N}\sum_{k=1}^{K}\Delta_{k,h}^{p} \leq 2\epsilon KN(H-h+1) + 4H\sqrt{\log(2KHN/\delta)}\sum_{k=1}^{K}\sum_{p=1}^{N}\sum_{\ell=h}^{H}\frac{1}{\sqrt{N_{k-1,\ell}(\gamma_{k\ell}^{p})}}\frac{1}{1+N_{k-1,\ell}(\gamma_{k\ell}^{p})}$$

$$+4\sum_{\ell=h}^{H}\sum_{p=1}^{N}\sum_{k=2}^{K}\frac{\sqrt{\beta_{k-1}\log(2KHN/\delta)}}{\sqrt{(N_{k-2,h}(\gamma_{k,h}^{p})+1)N_{k-1,h}(\gamma_{k,h}^{p})}}\frac{1}{1+N_{k-1,\ell}(\gamma_{k,\ell}^{p})}.$$

$$(33)$$

$\square$

**Lemma 11.** *Suppose $\mathcal{E}(\gamma_{kh}^{p})$ and $\mathcal{G}(\gamma_{kh}^{p})$ hold for all $\gamma_{kh}^{p}\in[\Gamma]$, then with probability $1-2\delta$, we have*

$$\sum_{p=1}^{N}\sum_{k=1}^{K}\hat{V}_{k,1}^{p}(s_{1}^{p})-V_{\pi^{kp}}^{\eta}(s_{1}^{p})$$
$$\leq 2\epsilon KHN + 8KH^{2}\sqrt{\Gamma N}\sqrt{\log(2KHN/\delta)} + 32H^{2}\sqrt{K\Gamma N}\sqrt{\log(HKN/\delta)}$$
$$+2KH^{5/2}\Gamma\sqrt{N}\sqrt{\log(2KH\Gamma)}\sqrt{\log(2KHN/\delta)}.$$

*Proof of Lemma 11.* Note that

$$\hat{V}_{k,h}^{p}(s_{k,h}^{p})-V_{h}^{\pi^{kp}}(s_{k,h}^{p})$$
$$= \hat{V}_{k,h}^{p}(s_{k,h}^{p})-Q_{h}^{*}(s_{k,h}^{p},a_{k,h}^{p})+Q_{h}^{*}(s_{k,h}^{p},a_{k,h}^{p})-V_{h}^{\pi^{kp}}(s_{k,h}^{p})$$
$$= \hat{Q}_{h}^{\pi^{kp}}(s_{k,h}^{p},a_{k,h}^{p})-Q_{h}^{*}(s_{k,h}^{p},a_{k,h}^{p})+Q_{h}^{*}(s_{k,h}^{p},a_{k,h}^{p})-Q_{h}^{\pi^{kp}}(s_{k,h}^{p},a_{k,h}^{p})$$
$$= \hat{Q}_{h}^{\pi^{kp}}(s_{k,h}^{p},a_{k,h}^{p})-Q_{h}^{*}(s_{k,h}^{p},a_{k,h}^{p})+P_{h}V_{h+1}^{*}(s_{k,h+1}^{p})-P_{h}V_{h+1}^{\pi^{kp}}(s_{k,h+1}^{p}) \qquad (34)$$
$$= [\hat{Q}_{h}^{\pi^{kp}}(s_{k,h}^{p},a_{k,h}^{p})-Q_{h}^{*}(s_{k,h}^{p},a_{k,h}^{p})]+[V_{h+1}^{*}(s_{k,h+1}^{p})-V_{h+1}^{\pi^{kp}}(s_{k,h+1}^{p})]$$
$$+[P_{h}V_{h+1}^{*}(s_{k,h+1}^{p})-V_{h+1}^{*}(s_{k,h+1}^{p})]+[V_{h+1}^{\pi^{kp}}(s_{k,h+1}^{p})-P_{h}V_{h+1}^{\pi^{kp}}(s_{k,h+1}^{p})]$$
$$= [\hat{Q}_{h}^{\pi^{kp}}(s_{k,h}^{p},a_{k,h}^{p})-Q_{h}^{*}(s_{k,h}^{p},a_{k,h}^{p})]+[\hat{V}_{k,h+1}^{\pi^{kp}}(s_{k,h+1}^{p})-V_{h+1}^{\pi^{kp}}(s_{k,h+1}^{p})]-\Delta_{k,h+1}^{p}$$
$$+[P_{h}V_{h+1}^{*}(s_{k,h+1}^{p})-V_{h+1}^{*}(s_{k,h+1}^{p})]+[V_{h+1}^{\pi^{kp}}(s_{k,h+1}^{p})-P_{h}V_{h+1}^{\pi^{kp}}(s_{k,h+1}^{p})].$$

Summing over $k=1,2,\ldots,K$, by (28), (31), (32) and (33) of Lemma 10 we have

$$\sum_{p=1}^{N}\sum_{k=1}^{K}\hat{V}_{k,h}^{p}(s_{k,h}^{p})-V_{h}^{\pi^{kp}}(s_{k,h}^{p})$$
$$\leq \sum_{p=1}^{N}\sum_{k=1}^{K}[\hat{V}_{k,h+1}^{p}(s_{k,h+1}^{p})-V_{h+1}^{\pi^{kp}}(s_{k,h+1}^{p})]-\sum_{p=1}^{N}\sum_{k=1}^{K}\Delta_{k,h+1}^{p}$$
$$+2KN\epsilon+4\sum_{p=1}^{N}\sum_{k=1}^{K}\frac{\alpha_{N_{k-1,h}(\gamma_{kh}^{p})}H\sqrt{\log(2KHN/\delta)}}{\sqrt{N_{k-1,h}(\gamma_{kh}^{p})}}$$
$$+2\sum_{p=1}^{N}\sum_{k=2}^{K}\frac{\alpha_{N_{k-1,h}(\gamma_{kh}^{p})}\sqrt{\beta_{k-1}\log(2KHN/\delta)}}{\sqrt{(N_{k-2,h}(\gamma_{kh}^{p})+1)N_{k-1,h}(\gamma_{kh}^{p})}}\frac{1}{1+N_{k-1,h}(\gamma_{kh}^{p})}+\sum_{k=1}^{K}\sum_{p=1}^{N}\Delta_{k,h+1}^{p} \qquad (35)$$
$$+\sum_{k=1}^{K}\sum_{p=1}^{N}[P_{h}V_{h+1}^{*}(s_{k,h+1}^{p})-V_{h+1}^{*}(s_{k,h+1}^{p})]$$
$$+\sum_{k=1}^{K}\sum_{p=1}^{N}[V_{h+1}^{\pi^{kp}}(s_{k,h+1}^{p})-P_{h}V_{h+1}^{\pi^{kp}}(s_{k,h+1}^{p})].$$

Note that (35) is recursive. Recall that $s_{k,1}^{p}=s_{1}^{p}$ for any $k\in[K], p\in[N]$. Thus by Lemma 10, when $\mathcal{E}(\gamma_{kh}^{p})$ and $\mathcal{G}(\gamma_{kh}^{p})$

hold for all $\gamma_{kh}^p$, for all $k \in [K], p \in [N]$, we have

$$
\sum_{p=1}^{N} \sum_{k=1}^{K} \hat{V}_{k,1}^p(s_1^p) - V_{\pi^{kp}}^\eta(s_1^p)
$$

$$
\leq 2\epsilon KHN + 4H\sqrt{\log(2KHN/\delta)} \sum_{k=1}^{K} \sum_{p=1}^{N} \sum_{h=1}^{H} \frac{1}{\sqrt{N_{k-1,h}(\gamma_{kh}^p)}} \frac{1}{1 + N_{k-1,h}(\gamma_{kh}^p)}
$$

$$
+ 4 \sum_{h=1}^{H} \sum_{p=1}^{N} \sum_{k=2}^{K} \frac{\sqrt{\beta_{k-1} \log(2KHN/\delta)}}{\sqrt{(N_{k-2,h}(\gamma_{kh}^p) + 1) N_{k-1,h}(\gamma_{kh}^p)}} \frac{1}{1 + N_{k-1,h}(\gamma_{kh}^p)}
$$

$$
+ \sum_{h=1}^{H} \sum_{k=1}^{K} \sum_{p=1}^{N} [V_{h+1}^{\pi^{kp}}(s_{k,h+1}^p) - P_h V_{h+1}^{\pi^{kp}}(s_{k,h}^p)].
$$

(36)

Further note that

$$
\sum_{k=1}^{K} \sum_{p=1}^{N} [P_h V_{h+1}^{\pi^{kp}}(s_{k,h+1}^p) - V_{h+1}^{\pi^{kp}}(s_{k,h+1}^p)]
$$

$$
= \sum_{\gamma \in [\Gamma]} \sum_{k=1}^{K} \sum_{p=1}^{N} \mathbf{1}\{\phi_h(s_{k,h}^p, a_{k,h}^p) = \gamma\} [P_h V_{h+1}^{\pi^{kp}}(s_{k,h+1}^p) - V_{h+1}^{\pi^{kp}}(s_{k,h+1}^p)]
$$

and

$$
\sum_{k=1}^{K} \sum_{p=1}^{N} [P_h V_{h+1}^*(s_{k,h+1}^p) - V_{h+1}^*(s_{k,h+1}^p)]
$$

$$
= \sum_{\gamma \in [\Gamma]} \sum_{k=1}^{K} \sum_{p=1}^{N} \mathbf{1}\{\phi_h(s_{k,h}^p, a_{k,h}^p) = \gamma\} [P_h V_{h+1}^*(s_{k,h+1}^p) - V_{h+1}^*(s_{k,h+1}^p)].
$$

By Azuma-Höeffding's inequality, with probability $\geq 1 - \delta/\Gamma$,

$$
\sum_{k=1}^{K} \sum_{p=1}^{N} \mathbf{1}\{\phi_h(s_{k,h}^p, a_{k,h}^p) = \gamma\} [P_h V_{h+1}^{\pi^{kp}}(s_{k,h+1}^p) - V_{h+1}^{\pi^{kp}}(s_{k,h+1}^p)] \leq 2H\sqrt{n_h^K(\gamma)} \sqrt{\log \frac{\Gamma}{\delta}},
$$

and with probability $\geq 1 - \delta/\Gamma$, we have

$$
\sum_{k=1}^{K} \sum_{p=1}^{N} \mathbf{1}\{\phi_h(s_{k,h}^p, a_{k,h}^p) = \gamma\} [P_h V_{h+1}^*(s_{k,h+1}^p) - V_{h+1}^*(s_{k,h+1}^p)] \leq 2H\sqrt{n_h^K(\gamma)} \sqrt{\log \frac{\Gamma}{\delta}},
$$

thus by summing over all the possible $\gamma_{kh}^p \in [\Gamma]$, with probability $1 - 2\delta$,

$$
\sum_{k=1}^{K} \sum_{p=1}^{N} [P_h V_{h+1}^{\pi^{kp}}(s_{k,h+1}^p) - V_{h+1}^{\pi^{kp}}(s_{k,h+1}^p)] + [P_h V_{h+1}^*(s_{k,h+1}^p) - V_{h+1}^*(s_{k,h+1}^p)]
$$

$$
\leq 4H \sum_{\gamma \in [\Gamma]} \sqrt{n_h^K(\gamma)} \sqrt{\log \frac{\Gamma}{\delta}}.
$$

Finally, note that

$$
\begin{aligned}
&\sum_{p=1}^{N}\sum_{k=1}^{K}\sum_{h=1}^{H}\frac{1}{\sqrt{N_{k-1,h}(\gamma_{kh}^{p})}}\frac{1}{1+N_{k-1,h}(\gamma_{kh}^{p})}\\
&\leq \sum_{h=1}^{H}\sum_{k=1}^{K}\sum_{\gamma\in[\Gamma]}\sum_{j=1}^{N_{k-1,h}(\gamma)}\frac{1}{j}\\
&\leq_{(1)}\sum_{k=1}^{K}\sum_{h=1}^{H}\sum_{\gamma\in[\Gamma]}\sqrt{N_{k-1,h}(\gamma)}\sqrt{\sum_{j=1}^{\infty}1/j^{2}}\leq_{(2)}\sum_{h=1}^{H}\sum_{k=1}^{K}\sum_{\gamma\in[\Gamma]}2\sqrt{N_{k-1,h}(\gamma)}\\
&\leq_{(3)}2\sqrt{HK\Gamma\sum_{h=1}^{H}\sum_{k=1}^{K}\sum_{\gamma\in[\Gamma]}N_{k-1,h}(\gamma)}\\
&=2HK\sqrt{\Gamma N}
\end{aligned}
\tag{37}
$$

where (1) and (3) hold by Cauchy's inequality, and (2) holds because $\sqrt{\pi^{2}/6}<2$.

Furthermore, we also have

$$
\sum_{h=1}^{H}\sum_{\gamma\in[\Gamma]}\sqrt{n_{h}^{K}(\gamma)}\leq\sqrt{H\Gamma\sum_{h=1}^{H}\sum_{\gamma\in[\Gamma]}n_{h}^{K}(\gamma)}=\sqrt{H^{2}K\Gamma N}.
\tag{38}
$$

Additionally, note that whenever $N_{k-2,h}(\gamma_{kh}^{p})\geq 1$, we have

$$
\begin{aligned}
&\sum_{h=1}^{H}\sum_{p=1}^{N}\sum_{k=2}^{K}\frac{\sqrt{\beta_{k-1}\log(2KHN/\delta)}}{\sqrt{(N_{k-2,h}(\gamma_{kh}^{p})+1)N_{k-1,h}(\gamma_{kh}^{p})}}\frac{1}{1+N_{k-1,h}(\gamma_{kh}^{p})}\\
&\leq\sum_{h=1}^{H}\sum_{p=1}^{N}\sum_{k=2}^{K}\frac{\sqrt{\beta_{k-1}\log(2KHN/\delta)}}{\sqrt{N_{k-2,h}(\gamma_{kh}^{p})+1}}\frac{1}{1+N_{k-1,h}(\gamma_{kh}^{p})},
\end{aligned}
\tag{39}
$$

and note that

$$
\begin{aligned}
&\sum_{h=1}^{H}\sum_{p=1}^{N}\sum_{k=2}^{K}\frac{1}{\sqrt{N_{k-2,h}(\gamma_{k+1,h}^{p})+1}}\frac{1}{1+N_{k-1,h}(\gamma_{k+1,h}^{p})}=_{(1)}\sum_{k=1}^{K}\sum_{h=1}^{H}\sum_{\gamma\in[\Gamma]}\frac{N_{k,h}(\gamma)}{\sqrt{N_{k-1,h}(\gamma)+1}}\frac{1}{1+N_{k,h}(\gamma)}\\
&\leq\sum_{k=1}^{K}\sum_{h=1}^{H}\sum_{\gamma\in[\Gamma]}\frac{1}{\sqrt{N_{k-1,h}(\gamma)}}\leq\sum_{k=1}^{K}\sum_{h=1}^{H}\sum_{\gamma\in[\Gamma]}\frac{\mathbf{1}\{N_{k,h}(\gamma\geq 1)\}}{\sqrt{N_{k,h}(\gamma)}}\leq\sqrt{KH\Gamma\sum_{k=1}^{K}\sum_{h=1}^{H}\sum_{\gamma\in[\Gamma]}N_{k,h}(\gamma)}=KH\Gamma\sqrt{N},
\end{aligned}
\tag{40}
$$

where equality (1) holds because $N_{k,h}(\gamma)$ is the number of agents that reach aggregated state $\gamma$ at period $h$ during episode $k$. And the last inequality holds by Cauchy's inequality. Recall that $\beta_{k}=\frac{1}{2}H^{3}\log(2kH\Gamma)$, thus by (39),

$$
\sum_{h=1}^{H}\sum_{p=1}^{N}\sum_{k=2}^{K}\frac{\sqrt{\beta_{k-1}\log(2KHN/\delta)}}{\sqrt{(N_{k-2,h}(\gamma_{kh}^{p})+1)N_{k-1,h}(\gamma_{kh}^{p})}}\frac{1}{1+N_{k-1,h}(\gamma_{kh}^{p})}\leq KH^{5/2}\Gamma\sqrt{N}\sqrt{\log(2KH\Gamma)}\sqrt{\log(2KHN/\delta)}.
\tag{41}
$$

Thus by (36), when events $\mathcal{E}(\gamma_{kh}^{p})$ and $\mathcal{G}(\gamma_{kh}^{p})$ hold for all $\gamma_{kh}^{p}\in[\Gamma]$, with probability $1-2\delta$, we have

$$
\begin{aligned}
&\sum_{p=1}^{N}\sum_{k=1}^{K}\hat{V}_{k,1}^{p}(s_{1}^{p})-V_{\pi^{k_{p}}}^{\eta}(s_{1}^{p})\\
&\leq 2\epsilon KHN+4KH^{2}\sqrt{\Gamma N}\sqrt{\log(2KHN/\delta)}+32H^{2}\sqrt{K\Gamma N}\sqrt{\log(HKN/\delta)}\\
&\quad+2KH^{5/2}\Gamma\sqrt{N}\sqrt{\log(2KH\Gamma)}\sqrt{\log(2KHN/\delta)}.
\end{aligned}
$$

$\square$

# D. Proofs for the Infinite-Horizon Case

**Additional notations**   For true MDP $M$ and policy $\pi$ we denote the discounted value function under $\pi$ at state $s$ as. $V_{\pi,1}^\eta(s) = V_\pi^\eta(s)$. We denote $V_*^\eta$ as the discounted value function under the optimal policy $\pi^*$.

During each pseudo-episode $k \in [K]$, each agent samples a random vector with independent components $w^{kp} \in \mathbb{R}^{H_k SA}$, where $w^{kp}(h, s, a) \sim \mathcal{N}(0, \sigma_k^2(h, s, a))$ and $\sigma_k(h, s, a) = \sqrt{\frac{\beta_k}{N_{k-1}(\phi(s,a))+1}}$, where $\beta_k$ is a tuning parameter, $N_{k-1,h}(\phi(s,a))$ is the total number of times that aggregated state $\phi(s, a)$ is reached across all agents during episode $k - 1$. Given $w^{kp}$, we construct a randomized perturbation of the empirical MDP for agent $p$ as

$$\overline{M}^{kp} = (T, \mathcal{S}, \mathcal{A}, \hat{P}^k, \hat{R}^k + w^{kp}, N), \tag{42}$$

where the empirical distributions $\hat{P}^k$ and empirical rewards $\hat{R}^k$ are computed as in (13) and (14). During each pseudo-episode $k \in [K]$, the data set $\tilde{D}_k^p$ contains perturbation of samples from pseudo-episode $k - 1$ used by agent $p$.

We define the following events:

$$\mathcal{E}^I(\gamma) := \left\{ \left| \frac{1}{N_{k-1}(\gamma)} \sum_{j=1}^N \sum_{h \in [H_{k-1}]} \mathbf{1}\{\phi(s_{k-1,h}^j, a_{k-1,h}^j) = \gamma\}\{V_*^\eta(s_{k-1,h}^j) - PV_*^\eta(s_{k-1,h}^j)\} \right| \right.$$
$$\left. \leq \frac{2 H_{k-1} \sqrt{\log(TN/\delta)}}{\sqrt{N_{k-1,h}(\gamma)}} \right\}. \tag{43}$$

$$\mathcal{G}^I(\gamma) := \left\{ \left| \frac{1}{N_{k-1}(\gamma)} \sum_{j=1}^N \sum_{h \in [H_{k-1}]} \mathbf{1}\{\phi(s_{k-1,h}^j, a_{k-1,h}^j) = \gamma\} \tilde{Q}_{k-1,h}^j(s_{k-1,h}^j, a_{k-1,h}^j) \right| \right.$$
$$\left. \leq \frac{2\sqrt{\beta_{k-1} \log(TN/\delta)}}{\sqrt{(N_{k-1}(\gamma) + 1) N_{k-1}(\gamma)}} \right\}. \tag{44}$$

**Lemma 12** (Lemma 2 of (Dong et al., 2022)). *For all $\pi, s \in \mathcal{S}$ and $\eta \in [0, 1)$, $\left| V_\pi^\eta(s) - \frac{\lambda_\pi(s)}{1-\eta} \right| \leq \tau_\pi$.*

**Remark 13.** *For weakly communicating $M$, the optimal average reward is state independent, so under Assumption 6, almost surely for any $s, s' \in \mathcal{S}$, we have $|V_*^\eta(s) - V_*^\eta(s')| \leq 2\tau_* \leq 2\tau$.*

**Regret Decomposition**   Recall that $\mathrm{Regret}(M, T, N, \mathrm{RLSVI}_{\beta,\alpha,\xi})$ denotes the regret under infinite-horizon case for MDP $M$. Here $K = \arg\max\{k : t_k \leq T\}$. We put $K$ explicitly here to derive the regret bound in a way dependent on the random $K$. Let $H_k$ be the length of pseudo-episode $k$. So we can decompose the regret as

$$\begin{aligned}
&\mathrm{Regret}(M, T, N, \mathrm{RLSVI}_{\beta,\alpha,\xi}) \\
&= \mathbb{E}_{K,H_k} \left[ \sum_{p=1}^N \sum_{k=1}^K \sum_{t=1}^{H_k} (\lambda_* - R_t^p) \big| M \right] \\
&= \mathbb{E}_{K,H_k} \left[ \sum_{p=1}^N \sum_{k=1}^K H_k \lambda_* - \sum_{p=1}^N \sum_{k=1}^K \sum_{t=1}^{H_k} R_t^p \big| M \right] \\
&= \underbrace{\mathbb{E}_{K,H_k} \left[ \sum_{p=1}^N \sum_{k=1}^K \{H_k \lambda_* - V_*^\eta(s_{k,1}^p)\} \big| M \right]}_{(a)} + \underbrace{\sum_{p=1}^N \sum_{k=1}^K \left\{ V_*^\eta(s_{k,1}^p) - V_{\pi^{kp}}^\eta(s_{k,1}^p) \right\}}_{(b)}
\end{aligned} \tag{45}$$

Recall from (11) that $V_*^\eta, V_{\pi^{kp}}^\eta \leq \frac{1}{1-\eta}$. Note that (a) in (45) is the difference between the optimal average reward weighted by the pseudo-horizons and the discounted reward, and part (b) is the sum of the differences between the optimal discounted value and cumulative the discounted value of the employed policies throughout $K$ pseudo-episodes by all agents. We provide an upper bound for the worst-case regret by bounding (a) and (b) respectively.

### D.1. Proof for Infinite-Horizon Main Result: Theorem 7

*Proof of Theorem 7.* By Azuma-Höeffding inequality, conditional on the trajectory during pseudo-episode $k-1$, with probability $1 - \delta/(2TN)$, we have

$$\left| \frac{1}{N_{k-1}(\gamma)} \sum_{j=1}^{N_{k-1}(\gamma)} \{V_*^\eta(s_{k-1,h+1}^j) - P_h V_*^\eta(s_{k-1,h}^j)\} \right| \le \frac{2\sqrt{\log(2TN/\delta)}}{(1-\eta)\sqrt{N_{k-1}(\gamma)}}.$$

Additionally, recall from Algorithm 4 that $\tilde{Q}_{k-1,h}^j(s_{k-1,h}^j, a_{k-1,h}^j) \sim \mathcal{N}\left(0, \frac{\beta_{k-1}}{N_{k-1}(\gamma)+1}\right)$. Note that conditional on the trajectories during pseudo-episode $k-1$, the random perturbations are i.i.d. across $j \in [p], h \in [H_{k-1}]$, thus by Höefdding's inequality, conditional on the trajectory during pseduo-period $k-1$, with probability $1 - \delta/(2TN)$, we have

$$\left| \frac{1}{N_{k-1}(\gamma)} \sum_{j=1}^{N} \sum_{h\in[H_{k-1}]} \mathbf{1}\{\phi_h(s_{k-1,h}^p, a_{k-1,h}^p) = \gamma\} \tilde{Q}_{k-1,h}^j(s_{k-1,h}^p, a_{k-1,h}^p) \right| \le 2\sqrt{\frac{\beta_{k-1}}{N_{k-1}(\gamma)+1}} \sqrt{\frac{\log(2TN/\delta)}{N_{k-1}(\gamma)}}.$$

So we have

$$\mathbb{P}(\mathcal{E}(\gamma_{kh}^p), \mathcal{G}(\gamma_{kh}^p), \gamma_{kh}^p, \forall k \in [K], h \in [H_k], p \in [N]) = \mathbb{P}(\mathcal{E}(\gamma_t^p), \mathcal{G}(\gamma_t^p), \gamma_t^p, \forall t \in [T], p \in [N])$$
$$\ge 1 - \sum_{t\in[T],p\in[N]} (\mathbb{P}(\mathcal{E}(\gamma_t^p)^c) + \mathbb{P}(\mathcal{G}(\gamma_t^p)^c)) \ge 1 - 2NT\delta/(2NT) \ge 1 - \delta. \tag{46}$$

By Lemma 15 and Lemma 14, with probability $1 - 3\delta$, we have

$$\begin{aligned}
\text{Regret}(M, T, N, \text{RLSVI}_{\beta,\alpha,\xi}) &\le \frac{\tau N[(1-\eta)T+1]}{\sqrt{NT}} + [(1-\eta)T+1]\tau\left(1 + \sqrt{N\log(NT)}\right) \\
&+ \frac{8T\sqrt{\Gamma N \log(2TN/\delta)}}{(1-\eta)^2} + \frac{8\Gamma\tau^{3/2}T\sqrt{N\log(2\tau\Gamma T)\log(2TN/\delta)}}{(1-\eta)^2} \\
&+ \frac{4K\sqrt{N\Gamma}\sqrt{\log(\Gamma/\delta)}}{1-\eta} + 2\eta\epsilon TN.
\end{aligned} \tag{47}$$

Recall that $\tau = \max\{\tau, \tau\}$, so with probability $1 - \delta$, we have

$$\begin{aligned}
\text{Regret}(M, T, N, \text{RLSVI}_{\beta,\alpha,\xi}) &\le \frac{\tau N[(1-\eta)T+1]}{\sqrt{NT}} + [(1-\eta)T+1]\tau\left(1 + \sqrt{N\log(NT)}\right) \\
&+ \frac{8T\sqrt{\Gamma N \log(6TN/\delta)}}{(1-\eta)^2} + \frac{8\tau^{3/2}T\Gamma\sqrt{N\log(2\tau\Gamma T)\log(6TN/\delta)}}{(1-\eta)^2} \\
&+ \frac{4K\sqrt{N\Gamma}\sqrt{\log(3\Gamma/\delta)}}{1-\eta} + 2\eta\epsilon TN.
\end{aligned}$$

When $1 - \eta$ such that $\frac{1}{(1-\eta)^2} \le C$ for some constant $C$, is bounded from below, we have

$$\begin{aligned}
\text{Regret}(M, T, N, \text{RLSVI}_{\beta,\alpha,\xi}) &\le 2\epsilon TN + 2\tau\sqrt{NT} + 4T\tau\sqrt{N\log(NT)} \\
&+ 16C\max\{\tau^{3/2}, 1\}T\Gamma\sqrt{\Gamma N\log(2\tau T)\log(6TN/\delta)}
\end{aligned}$$

$\square$

### D.2. Lemmas for bounding (a) and (b) in (45)

**Lemma 14** (Bound for (a) of (45)). $\mathbb{E}_{H_k,K}\left[\sum_{p=1}^N \sum_{k=1}^K H_k\lambda_* - V_*^\eta(s_{k,1}^p)\right] \le \frac{\tau N[(1-\eta)T+1]}{\sqrt{NT}} + [(1-\eta)T+1]\tau\left(1 + \sqrt{N\log(NT)}\right).$

*Proof of Lemma 14.* Note that the expected length of each pseudo-episode is independent of the policy and is equal to $\frac{1}{1-\eta}$. Thus for $K$ fixed, $\mathbb{E}_{H_k}\left[\sum_{p=1}^{N}\sum_{k=1}^{K}H_k\lambda_* - V_*^{\eta}(s_{k,1}^{p})\right] = \sum_{p=1}^{N}\sum_{k=1}^{K}\left(\frac{\lambda_*}{1-\eta} - V_*^{\eta}(s_{k,1}^{p})\right)$, thus

$$\mathbb{E}\left[\sum_{p=1}^{N}\sum_{k=1}^{K}H_k\lambda_* - V_*^{\eta}(s_{k,1}^{p})\right] \le \mathbb{E}\left[\sum_{p=1}^{N}\sum_{k=1}^{K}\left|\frac{\lambda_*}{1-\eta} - V_*^{\eta}(s_{k,1}^{p})\right|\right].$$

By Lemma 12, we know that $\left|\frac{\lambda_*}{1-\eta} - V_*^{\eta}(s_{k,1}^{p})\right| \le \tau$. So that for any $p \in [N]$, for any fixed $K$,

$$\sum_{k=1}^{K}\left|\frac{\lambda_*}{1-\eta} - V_*^{\eta}(s_{k,1}^{p})\right| \le K\tau.$$

Note that $s_{k,1}^{p}$ are sampled i.i.d. across $p$ at the beginning of each pseudo-episode $k$. Höeffding's inequality shows that for any $\epsilon > 0$,

$$\mathbb{P}\left(\sum_{p=1}^{N}\sum_{k=1}^{K}\left|\frac{\lambda_*}{1-\eta} - V_*^{\eta}(s_{k,1}^{p})\right| \ge \epsilon + K\tau\right) \le \exp\left(-\frac{2\epsilon^2}{4NK^2\tau^2}\right).$$

Take

$$\epsilon = K\tau\sqrt{N\log(NT)},$$

we then have

$$\mathbb{P}\left(\sum_{p=1}^{N}\sum_{k=1}^{K}\left|\frac{\lambda_*}{1-\eta} - V_*^{\eta}(s_{k,1}^{p})\right| \ge K\tau\left(1 + \sqrt{N\log(NT)}\right)\right) \le \frac{1}{\sqrt{NT}}.$$

Thus conditioning on the total number of pseudo-episodes $K$,

$$\mathbb{E}_K\left[\sum_{p=1}^{N}\sum_{k=1}^{K}\left|\frac{\lambda_*}{1-\eta} - V_*^{\eta}(s_{k,1}^{p})\right|\right] \le \frac{\tau KN}{\sqrt{NT}} + K\tau\left(1 + \sqrt{N\log(NT)}\right).$$

Further note that since $H_k \sim \text{Geometric}(1-\eta)$ and are i.i.d. across $k$, so

$$\mathbb{E}[K] \le (1-\eta)T + 1.$$

Hence by taking expectation over $K$, we have

$$\mathbb{E}\left[\sum_{p=1}^{N}\sum_{k=1}^{K}\left|\frac{\lambda_*}{1-\eta} - V_*^{\eta}(s_{k,1}^{p})\right|\right] \le \frac{\tau\mathbb{E}[K]N}{\sqrt{NT}} + \mathbb{E}[K]\tau\left(1 + \sqrt{N\log(NT)}\right)$$

$$\le \frac{\tau N[(1-\eta)T + 1]}{\sqrt{NT}} + [(1-\eta)T + 1]\tau\left(1 + \sqrt{N\log(NT)}\right).$$

$\square$

**Lemma 15** (Bound for (b) of (45)). *Suppose events $\mathcal{E}^I(\gamma)$ and $\mathcal{G}^I(\gamma)$ hold for any $\gamma \in [\Gamma]$, then with probability $1 - 2\delta$, we have*

$$\sum_{p=1}^{N}\sum_{k=1}^{K}\left\{V_*^{\eta}(s_{k,1}^{p}) - V_{\pi^{k_P},M}^{\eta}(s_{k,1}^{p})\right\} \le \frac{8T\sqrt{\Gamma N\log(2TN/\delta)}}{(1-\eta)^2} + \frac{8\tau^{3/2}T\Gamma\sqrt{N\log(2\tau\Gamma T)\log(2TN/\delta)}}{1-\eta}$$

$$+ \frac{4T\sqrt{N\Gamma}\sqrt{\log(\Gamma/\delta)}}{1-\eta} + 2\eta\epsilon TN.$$

*Proof of Lemma 15.* Recall from Algorithm 4 that the unclipped value function estimates $\bar{Q}^p_{k,h}(\cdot)$ during pseudo-episode $k$ at time period $h \in [H_k]$ (recall that $H_k$ here is random) as

$$\bar{Q}^p_{k,h}(\gamma) = \arg\min_{Q \in \mathbb{R}} \sum_{(s,a):\phi_h(s,a)=\gamma} (Q - \xi_{N_{k-1}(\gamma)} - (1 - \alpha_{N_{k-1}(\gamma)})\hat{Q}_{k-1}(\gamma)$$
$$-\eta\alpha_{N_{k-1,h}(\gamma)}\{r(s,a) + \max_{a' \in \mathcal{A}} \hat{Q}^p_{k,h+1}(s',a')\})^2 + \left\| Q - \eta\alpha_{N_{k-1,h}(\gamma)}\tilde{Q}^p_{k,h} \right\|^2_2.$$

So similar to the derivation in the proof of Lemma 10, we have

$$\bar{Q}^p_{k,h}(\gamma) = \xi_{N_{k-1}(\gamma)} + \frac{1}{N_{k-1}(\gamma)} \sum_{p=1}^{N_{k-1}(\gamma)} (1 - \alpha_{N_{k-1}(\gamma)})\hat{Q}^p_{k-1}(\gamma)$$
$$+\alpha_{N_{k-1}(\gamma)}\eta(r(s^p_{k-1}, a^p_{k-1}) + \hat{V}^p_k(s^p_{k-1,h+1}) + \tilde{Q}^p_{k,h}(s^p_{k-1,h}, a^p_{k-1,h})).$$

By definition, we have $\bar{Q}^p_{k,1}(\gamma) = \bar{Q}^p_{t_k}(\gamma)$. We denote $\bar{Q}^p_k(\gamma) := \bar{Q}^p_{k,1}(\gamma)$ in the following. Thus for any $\gamma = \phi(s^p_{k,h}, a^p_{k,h})$ during pseudo-episode $k$, with $\phi(s^j_{k-1,h}, a^j_{k-1,h}) = \gamma$ in the following, where $\{(s^j_{k-1,h}, a^j_{k-1,h})\}$ come from all the state-action pairs during pseudo-period $k - 1$ (note that we don't distinguish between different pseudo periods now), where $h \in \{1, 2, \ldots, H_{k-1}\}$, and $H_{k-1}$ is the random length of pseudo-episode $k - 1$, and recall that $N_{k-1}(\gamma) = \sum_{p=1}^N \sum_{h \in [H_{k-1}]} \mathbf{1}\{\phi_h(s^p_{k-1,h}, a^p_{k-1,h}) = \gamma\}$, so we have

$$\bar{Q}^p_{k,h}(s^p_{k,h}, a^p_{k,h}) - Q^\eta_*(s^p_{k,h}, a^p_{k,h})$$
$$= \eta\xi_{N_{k-1}(\gamma)} + \frac{1}{N_{k-1}(\gamma)} \sum_{j=1}^{N_{k-1}(\gamma)} (1 - \alpha_{N_{k-1}(\gamma)})(Q^\eta_*(s^p_{k,h}, a^p_{k,h}) - \hat{Q}^j_{k-1}(s^j_{k-1,h}, a^j_{k-1,h}))$$
$$+\eta\frac{\alpha_{N_{k-1}(\gamma)}}{N_{k-1}(\gamma)} \sum_{p=1}^{N_{k-1}(\gamma)} \{-[r^j_{k-1,h}(s^j_{k-1,h}, a^j_{k-1,h}) + \hat{V}^j_{k-1}(s^j_{k-1,h+1}) + \tilde{Q}^j_{k-1,h}(s^j_{k-1,h}, a^j_{k-1,h})]$$
$$+Q^\eta_*(s^j_{k-1,h}, a^j_{k-1,h})\}$$
$$+\frac{\alpha_{N_{k-1,h}(\gamma)}}{N_{k-1}(\gamma)} \sum_{j=1}^{N_{k-1,h}(\gamma)} \underbrace{\{Q^\eta_*(s^p_{k,h}, a^p_{k,h}) - Q^\eta_*(s^j_{k-1,h}, a^j_{k-1,h}))\}}_{\le \epsilon}$$
$$\le \epsilon + \eta\xi_{N_{k-1}(\gamma)} + \frac{\eta}{N_{k-1}(\gamma)} \sum_{j=1}^{N_{k-1}(\gamma)} (1 - \alpha_{N_{k-1}(\gamma)})(Q^\eta_*(s^p_{k,h}, a^p_{k,h}) - \hat{Q}^j_{k-1}(s^j_{k-1,h}, a^j_{k-1,h})) \quad (48)$$
$$+\eta\frac{\alpha_{N_{k-1}(\gamma)}}{N_{k-1,h}(\gamma)} \sum_{p=1}^{N_{k-1}(\gamma)} \{-[r^j_{k-1,h}(s^j_{k-1,h}, a^j_{k-1,h}) + \hat{V}^j_{k-1}(s^j_{k-1,h+1}) + \tilde{Q}^j_{k-1,h}(s^j_{k-1,h}, a^j_{k-1,h})]$$
$$= \epsilon + \eta\xi_{N_{k-1}(\gamma)} - \eta\frac{\alpha_{N_{k-1}(\gamma)}}{N_{k-1}(\gamma)} \sum_{j=1}^{N_{k-1}(\gamma)} \tilde{Q}^j_{k-1,h}(s^j_{k-1,h}, a^j_{k-1,h})$$
$$-\frac{\eta}{N_{k-1}(\gamma)} \sum_{j=1}^{N_{k-1}(\gamma)} \{\hat{V}^j_{k-1}(s^j_{k-1,h+1}) - V^\eta_*(s^j_{k-1,h+1})\}$$
$$-\eta\frac{\alpha_{N_{k-1}(\gamma)}}{N_{k-1}(\gamma)} \sum_{j=1}^{N_{k-1}(\gamma)} \{V^\eta_*(s^j_{k-1,h+1}) - P_h V^\eta_*(s^j_{k-1,h})\},$$

where we used the fact that under optimal policy $\pi^*$ for the true MDP, we have

$$Q_*(s', a') = r(s', a') + \eta P_* V_*(s'),$$

and under the optimal policy $\pi^{k-1,j}$ under MDP $\overline{M}^{k-1,j}$, we have

$$\hat{Q}^j_{k-1}(s', a') = r(s', a') + \eta\hat{P}_{\pi^{kj}}\hat{V}_{k-1}(s).$$

Now suppose that events $\mathcal{E}^I(\gamma^p_{kh})$ and $\mathcal{G}^I(\gamma^p_{kh})$ hold for all aggregated states $\gamma^p_{kh}$ during pseudo-perido $k$ for all $k \in [K]$. Then by similar derivation as that for the finite-horion case in Lemma 9, we have

$$\hat{V}^p_{k,h}(s) - V_*(s) \ge 0, \ \forall s \in \mathcal{S}, k \in [K], p \in [N]. \quad (49)$$

We denote

$$\Delta_{k,h}^p = \hat{V}_{k,h}^p(s_{k,h}^p) - V_*^\eta(s_{k,h}^p).$$

Note that

$$\hat{V}_{k,h}(s_{k,h}^p) - V_*(s_{k,h}^p) \le \hat{Q}_{k,h}^p(s_{k,h}^p, a_{k,h}^p) - Q_*^\eta(s_{k,h}^p, a_{k,h}^p) \le \bar{Q}_{k,h}^p(s_{k,h}^p, a_{k,h}^p) - Q_*^\eta(s_{k,h}^p, a_{k,h}^p).$$

By (48) we have

$$
\begin{aligned}
&\bar{Q}_{k,h}^p(s_{k,h}^p, a_{k,h}^p) - Q_*^\eta(s_{k,h}^p, a_{k,h}^p) \\
&= \eta \xi_{N_{k-1}(\gamma)} + \frac{\eta}{N_{k-1}(\gamma)} \sum_{j=1}^{N_{k-1}(\gamma)} (1 - \alpha_{N_{k-1}(\gamma)})(Q_*^\eta(s_{k,h}^p, a_{k,h}^p) - \hat{Q}_{k-1}^j(s_{k-1,h}^j, a_{k-1,h}^j)) \\
&\quad + \eta \frac{\alpha_{N_{k-1}(\gamma)}}{N_{k-1}(\gamma)} \sum_{p=1}^{N_{k-1}(\gamma)} \{ -[r_{k-1,h}^j(s_{k-1,h}^j, a_{k-1,h}^j) + \hat{V}_{k-1}^j(s_{k-1,h+1}^j) + \tilde{Q}_{k-1,h}^j(s_{k-1,h}^j, a_{k-1,h}^j)] \\
&\qquad\qquad\qquad\qquad + Q_*^\eta(s_{k-1,h}^j, a_{k-1,h}^j) \} \\
&\quad + \eta \frac{\alpha_{N_{k-1,h}(\gamma)}}{N_{k-1}(\gamma)} \sum_{j=1}^{N_{k-1,h}(\gamma)} \underbrace{\{ Q_*^\eta(s_{k,h}^p, a_{k,h}^p) - Q_*^\eta(s_{k-1,h}^j, a_{k-1,h}^j)) \}}_{\le \epsilon} \\
&\le \eta\epsilon + \xi_{N_{k-1}(\gamma)} + \frac{1}{N_{k-1}(\gamma)} \sum_{j=1}^{N_{k-1}(\gamma)} (1 - \alpha_{N_{k-1}(\gamma)})(Q_*^\eta(s_{k,h}^p, a_{k,h}^p) - \hat{Q}_{k-1}^j(s_{k-1,h}^j, a_{k-1,h}^j)) \qquad (50) \\
&\quad + \eta \frac{\alpha_{N_{k-1}(\gamma)}}{N_{k-1,h}(\gamma)} \sum_{p=1}^{N_{k-1}(\gamma)} \{ -[r_{k-1,h}^j(s_{k-1,h}^j, a_{k-1,h}^j) + \hat{V}_{k-1}^j(s_{k-1,h+1}^j) + \tilde{Q}_{k-1,h}^j(s_{k-1,h}^j, a_{k-1,h}^j)] \\
&= \eta \xi_{N_{k-1}(\gamma)} + \eta\epsilon - \eta \frac{\alpha_{N_{k-1}(\gamma)}}{N_{k-1}(\gamma)} \sum_{j=1}^{N_{k-1}(\gamma)} \tilde{Q}_{k-1,h}^j(s_{k-1,h}^j, a_{k-1,h}^j) \\
&\quad - \frac{\eta}{N_{k-1}(\gamma)} \sum_{j=1}^{N_{k-1}(\gamma)} \{ \hat{V}_{k-1}^j(s_{k-1,h+1}^j) - V_*^\eta(s_{k-1,h+1}^j) \} \\
&\quad - \eta \frac{\alpha_{N_{k-1}(\gamma)}}{N_{k-1}(\gamma)} \sum_{j=1}^{N_{k-1}(\gamma)} \{ V_*^\eta(s_{k-1,h+1}^j) - P_h V_*^\eta(s_{k-1,h}^j) \},
\end{aligned}
$$

where the inequality follows by definition of $\epsilon$-error aggregated states as in Definition 4.

Then by similar derivation as for (32), by summing from the $i$-th pseudo-episode to $K$-th pseudo-episode, we have

$$
\begin{aligned}
\sum_{p=1}^N \sum_{k=i}^K \Delta_{k,h}^p \quad &\le 2N\eta \sum_{k=i}^K \epsilon + \frac{4\eta}{1-\eta} \sum_{p=1}^N \sum_{k=i}^K \frac{\alpha_{N_{k-1,h}(\gamma_{kh}^p)} \sqrt{\log(2TN/\delta)}}{\sqrt{N_{k-1,h}(\gamma_{kh}^p)}} \\
&\quad + \eta \sum_{p=1}^N \sum_{k=i+1}^K \frac{\alpha_{N_{k-1,h}(\gamma_{kh}^p)}}{N_{k-1,h}(\gamma_{kh}^p)} \sum_{j=1}^{N_{k-1,h}(\gamma_{kh}^p)} \Delta_{k,h+1}^j.
\end{aligned}
$$

Then by similar derivation as in (33) under events $\mathcal{E}^I(\gamma_{kh}^p)$ and $\mathcal{G}^I(\gamma_{kh}^p)$ for all $\gamma_{kh}^p$ across $K$ pseudo-episodes and $p$ agents, we have

$$
\begin{aligned}
\sum_{p=1}^N \sum_{k=i}^K \Delta_{k,h}^p \quad &\le 2\eta\epsilon N \sum_{k=i}^K H_k + \frac{4\eta}{1-\eta} \sqrt{\log(2TN/\delta)} \sum_{k=i}^K \sum_{p=1}^N \sum_{\ell=h}^H \frac{1}{\sqrt{N_{k-1,\ell}(\gamma_{k\ell}^p)}} \frac{1}{1 + N_{k-1,\ell}(\gamma_{k\ell}^p)} \\
&\quad + 4\eta \sum_{\ell=h}^H \sum_{p=1}^N \sum_{k=i+1}^K \frac{\sqrt{\beta_{k-1} \log(2TN/\delta)}}{\sqrt{(N_{k-2,h}(\gamma_{k,h}^p) + 1) N_{k-1,h}(\gamma_{k,h}^p)}} \frac{1}{1 + N_{k-1,\ell}(\gamma_{k,\ell}^p)}.
\end{aligned}
$$

Then following similar steps as in the proof of Lemma 11, we have $\Delta_{k,h}^p = \hat{V}_{k,h}^p(s_{k,h}^p) - V_*^\eta(s_{k,h}^p)$.

$$
\begin{aligned}
&\hat{V}_{k,h}^p(s_{k,h}^p) - V_{\pi^{kp}}^\eta(s_{k,h}^p) \\
&= [\hat{Q}_h^{\pi^{kp}}(s_{k,h}^p, a_{k,h}^p) - Q_*^\eta(s_{k,h}^p, a_{k,h}^p)] + \eta[\hat{V}_{k,h+1}^{\pi^{kp}}(s_{k,h+1}^p) - V_{\pi^{kp}}^\eta(s_{k,h+1}^p)] - \eta\Delta_{k,h+1}^p \\
&\quad + \eta[V_{\pi^{kp}}^\eta(s_{k,h+1}^p) - P_h V_{\pi^{kp}}^\eta(s_{k,h+1}^p)].
\end{aligned} \tag{51}
$$

Note that (51) is recursive, so when events $\mathcal{E}^I(\gamma_{kh}^p)$ and $\mathcal{G}^I(\gamma_{kh}^p)$ hold for all aggregated states $\gamma_{kh}^p$ during pseudo-perido $k$ for all $k \in [K]$, we have

$$
\begin{aligned}
&\sum_{k=1}^K \sum_{p=1}^N \hat{V}_{k,1}^p(s_1^p) - V_{\pi^{kp}}^\eta(s_1^p) \\
&\leq 2\eta\epsilon TN + \frac{4}{1-\eta}\sqrt{\log(2TN/\delta)} \sum_{k=1}^K \sum_{p=1}^N \sum_{h=1}^{H_k} \eta^{h-1} \frac{1}{\sqrt{N_{k-1}(\gamma_{kh}^p)}} \frac{1}{1+N_{k-1}(\gamma_{kh}^p)} \\
&\quad + 4\sum_{k=2}^K \sum_{h=1}^{H_k} \eta^{h-1} \sum_{p=1}^N \frac{\sqrt{\beta_{k-1}\log(2TN/\delta)}}{\sqrt{(N_{k-2}(\gamma_{kh}^p)+1)N_{k-1}(\gamma_{kh}^p)}} \frac{1}{1+N_{k-1}(\gamma_{kh}^p)} \\
&\quad + \sum_{k=1}^K \sum_{h=1}^{H_k} \eta^{h-1} \sum_{p=1}^N [P_h V_*^\eta(s_{k,h+1}^p) - V_*^\eta(s_{k,h+1}^p)] \\
&\quad + \sum_{k=1}^K \sum_{h=1}^{H_k} \eta^{h-1} \sum_{p=1}^N [V_{\pi^{kp}}^\eta(s_{k,h+1}^p) - P_{\pi^{kp}}^h V_{\pi^{kp}}^\eta(s_{k,h+1}^p)].
\end{aligned} \tag{52}
$$

Note that

$$
\begin{aligned}
&\frac{4}{1-\eta}\sqrt{\log(2TN/\delta)} \sum_{k=1}^K \sum_{p=1}^N \sum_{h=1}^{H_k} \eta^{h-1} \frac{1}{\sqrt{N_{k-1}(\gamma_{kh}^p)}} \frac{1}{1+N_{k-1}(\gamma_{kh}^p)} \\
&\leq \frac{4}{1-\eta}\sqrt{\log(2TN/\delta)} \sum_{k=1}^K \sum_{h=1}^{H_k} \eta^{h-1} \sum_{p=1}^N \frac{1}{\sqrt{N_{k-1}(\gamma_{kh}^p)}} \frac{1}{1+N_{k-1}(\gamma_{kh}^p)} \\
&\leq \frac{4}{1-\eta}\sqrt{\log(2TN/\delta)} \sum_{k=1}^K \sum_{h=1}^{H_k} \eta^{h-1} \sum_{\gamma\in[\Gamma]} \sum_{j=1}^{N_{k-1}(\gamma)} 1/j \\
&\leq \frac{4}{1-\eta}\sqrt{\log(2TN/\delta)} \sum_{k=1}^K \sum_{h=1}^{H_k} \eta^{h-1} \sum_{\gamma\in[\Gamma]} 2\sqrt{N_{k-1}(\gamma)} \\
&\leq \frac{8}{(1-\eta)}\sqrt{\log(2TN/\delta)} \sum_{k=1}^K \sum_{h=1}^{H_k} \eta^{h-1} \sqrt{\Gamma \sum_{\gamma\in[\Gamma]} N_{k-1}(\gamma)} \\
&\leq \frac{8}{(1-\eta)}\sqrt{\log(2TN/\delta)} \sum_{k=1}^K \sum_{h=1}^{H_k} \eta^{h-1} \sqrt{\Gamma N H_{k-1}} \leq \frac{8\sqrt{\Gamma N \log(2TN/\delta)}}{(1-\eta)} \sum_{h=1}^\infty \eta^{h-1} \sum_{k=1}^K \sqrt{H_{k-1}} \\
&\leq \frac{8\sqrt{\Gamma N \log(2TN/\delta)}}{(1-\eta)^2} \sum_{k=1}^K H_{k-1} \leq \frac{8T\sqrt{\Gamma N \log(2TN/\delta)}}{(1-\eta)^2}.
\end{aligned} \tag{53}
$$

Next, note that $\beta_k = \frac{1}{2}\tau^3 \log(2\tau\Gamma k)$, following similar steps as (53) and (40) we have

$$
4\sum_{k=1}^K \sum_{h=1}^{H_k} \eta^{h-1} \sum_{p=1}^N \frac{\sqrt{\beta_{k-1}\log(2TN/\delta)}}{\sqrt{(N_{k-2}(\gamma_{kh}^p)+1)N_{k-1}(\gamma_{kh}^p)}} \frac{1}{1+N_{k-1}(\gamma_{kh}^p)} \leq \frac{8\tau^{3/2}T\Gamma\sqrt{N\log(2\tau\Gamma T)\log(2TN/\delta)}}{(1-\eta)^2}. \tag{54}
$$

Additionally, denote $N_{k,h}(\gamma)$ as the number of times that aggregated state $\gamma$ is attained during pseudo-episode $k$ and pseudo

period $h$, then by Azuma-Höeffding's inequality, with probability $1 - 2\delta$, we have

$$\sum_{p=1}^{N} \{[V_{\pi^{k_p}}^{\eta}(s_{k,h+1}^p) - P_{\pi^{k_p}}^h V_{\pi^{k_p}}^{\eta}(s_{k,h+1}^p)] + [P_h V_*^{\eta}(s_{k,h+1}^p) - V_*^{\eta}(s_{k,h+1}^p)]\} \leq \frac{4}{1-\eta} \sum_{\gamma \in [\Gamma]} \sqrt{N_{k,h}(\gamma)} \sqrt{\log \frac{\Gamma}{\delta}}.$$

Hence with probability $1 - 2\delta$,

$$\begin{aligned}
&\sum_{k=1}^{K} \sum_{h=1}^{H_k} \eta^{h-1} \sum_{p=1}^{N} \{[P_h V_*^{\eta}(s_{k,h+1}^p) - V_*^{\eta}(s_{k,h+1}^p)] + [V_{\pi^{k_p}}^{\eta}(s_{k,h+1}^p) - P_{\pi^{k_p}}^h V_{\pi^{k_p}}^{\eta}(s_{k,h+1}^p)]\} \\
&\leq \frac{4\sqrt{\log(\Gamma/\delta)}}{1-\eta} \sum_{k=1}^{K} \sum_{h=1}^{H_k} \eta^{h-1} \sum_{\gamma \in [\Gamma]} \sqrt{N_k(\gamma)} \leq \frac{4\sqrt{\Gamma N \log(\Gamma/\delta)}}{1-\eta} \sum_{k=1}^{K} \sum_{h=1}^{H_k} \eta^{h-1} \\
&\leq \frac{4K\sqrt{\Gamma N \log(\Gamma/\delta)}}{(1-\eta)^2}
\end{aligned} \tag{55}$$

Thus by (52), (53), (54), (55), when events $\mathcal{E}^I(\gamma_{kh}^p)$ and $\mathcal{G}^I(\gamma_{kh}^p)$ hold for all aggregated states $\gamma_{kh}^p$ during pseudo-perido $k$ for all $k \in [K]$, with probability $1 - 2\delta$, we have

$$\begin{aligned}
\sum_{k=1}^{K} \sum_{p=1}^{N} \hat{V}_{k,1}^p(s_1^p) - V_{\pi^{k_p}}^{\eta}(s_1^p) \quad &\leq \frac{8T\sqrt{\Gamma N \log(2TN/\delta)}}{(1-\eta)^2} + \frac{8\tau^{3/2}T\Gamma\sqrt{N \log(2\tau\Gamma T) \log(2TN/\delta)}}{(1-\eta)^2} \\
&\quad + \frac{4K\sqrt{\Gamma N \log(\Gamma/\delta)}}{(1-\eta)^2}.
\end{aligned} \tag{56}$$

Finall, by (49), when events $\mathcal{E}^I(\gamma_{kh}^p)$ and $\mathcal{G}^I(\gamma_{kh}^p)$ hold for all aggregated states $\gamma_{kh}^p$ during pseudo-perido $k$ for all $k \in [K]$,

$$\hat{V}_{k,h}^p(s) - V_*(s) \geq 0, \ \forall s \in \mathcal{S}, k \in [K], p \in [N],$$

And with probability $1 - 2\delta$, we have

$$\begin{aligned}
\sum_{p=1}^{N} \sum_{k=1}^{K} \left\{ V_*^{\eta}(s_{k,1}^p) - V_{\pi^{k_p},M}^{\eta}(s_{k,1}^p) \right\} \quad &\leq \frac{8T\sqrt{\Gamma N \log(2TN/\delta)}}{(1-\eta)^2} + \frac{8\tau^{3/2}T\Gamma\sqrt{N \log(2\tau\Gamma T) \log(2TN/\delta)}}{1-\eta} \\
&\quad + \frac{4T\sqrt{N\Gamma}\sqrt{\log(\Gamma/\delta)}}{1-\eta}.
\end{aligned}$$

$\square$

