# OpenReview forum: "Concurrent Reinforcement Learning with Aggregated States via  Randomized Least Squares Value Iteration"
_ICML.cc/2025/Conference — ICML 2025 poster_

### Official Review · Reviewer_35te · 2025-03-09

**Overall Recommendation:** 4

**Summary:**

The authors extend the classic Least Squares Value Iteration (LSVI) method to the setting of multi-agent systems, and show guarantees for a randomized variant.

**Claims And Evidence:**

Yes.

**Essential References Not Discussed:**

No.

**Experimental Designs Or Analyses:**

Yes, to some extent.

**Methods And Evaluation Criteria:**

Yes.

**Other Comments Or Suggestions:**

None.

**Other Strengths And Weaknesses:**

None.

The writing is clear!

**Questions For Authors:**

Please put an Impact Statement :)

Page 2:
Line 94:
Can you please elaborate on the difference of the techniques used?

Page 3:
Why is $V_h^\pi(s)$ defined in this way?

Page 4:
Can you please further explain $\hat R_{h, s, a}^{k, \texttt{full}}$ and $\hat P_{h, s, a}^{k, \texttt{full}}(s')$?

Page 4:
Can you please elaborate on the concept of aggregated-state representations?

Page 5:
I do not understand the perturbation that is captured by Equation (4).

Page 6:
Can you please clarify the choice of Equation (7)?

Page 8:
More discussion on future work is required! :)

**Relation To Broader Scientific Literature:**

The theoretical guarantees cover an important gap in the literature.

**Theoretical Claims:**

Yes, to some extent.

---

> ### Author Rebuttal · Authors · 2025-03-30
>
> We thank the reviewer for your positive feedback!
>
> 1. We will add an impact statement later :)
>
> 2. [2] improves the regret bound of [1] by using clipping in their algorithm to avoid unreasonable estimates of the value functions. The rest of their algorithm closely follows [1], using a similar proof approach: constructing a confidence set for the empirical MDP to bound its deviation from the true MDP. They decompose regret similarly to [1] and get a tighter upper bound for the approximation error of the $Q$-value function due to their clipping technique.
>
>      Our paper uses a different algorithm from [2] and proof strategy to improve the regret bound of [1]. Firstly, we do information aggregation at the aggregated-state level, unlike the algorithms in [1,2] that record the information for every pair of $(s,a)$. Secondly, we use a different loss function and penalty term for updating the estimates of $Q$-value functions. By the first-order condition the estimated $Q$-value function is updated according to the derivation between line 1004 and line 1028, $\alpha$ and $\xi$ are carefully chosen based on concentration lemma 8, so that (i) Lemma 9 holds, which is a key step to the proof of Lemma 10 using iteration trick; (ii) the term in (37) between line 1375 and line 1389 can be controlled by a tight upper bound using Cauchy-Schwarz inequality.
>
> 3. $V_h^{\pi}(s)$ is the value function of adopting policy $\pi$ at state s during period h, which is a standard definition (see e.g. [3]). We include $h$ in the subscript because we consider inhomogeneous transition probabilities and reward functions (i.e. the transition probabilities and reward functions are allowed to be different at each period $h$ in our model).
>
> 4. On the one hand, $R_{h,s,a}^{k,\textrm{full}}$ and $\hat P_{h,s,a}^{k,\textrm{full}}(s')$
> are estimates of rewards $R_{h,s,a}$ and transition probability $P_{h,s,a}(s')$  used in episode $k$ based upon all the historical data across all agents from episode $0$ to $k-1$, where $R_{h,s,a}$ is the reward received by state-action pair $(s,a)$ at time step $h$, and $P_{h,s,a}(s')$ is the probability of transferring to state $s'$ if taking action $a$ at state $s$ during time step $h$.  On the other hand, $\hat P_{h,s,a}^k$ and $\hat R_{h,s,a}^{k}$ only uses data in episode $k-1$ (much less data compared to the previous two estimates). The algorithm utilizing $R_{h,s,a}^{k,\textrm{full}}$ and $\hat{P}_{h,s,a}^{k,\textrm{full}}(s')$ has a lower regret bound because we use more data in the buffer (information set), but can be computationally infeasible especially as the number of agents gets large, as we explained in the paper.
>
> 5. As we pointed out at the beginning of subsection 2.1, when $SA$ is too large, the algorithm can be computationally infeasible. So we aggregate the state-action pairs that have close $Q$-values under optimal policy. By $\epsilon$-error aggregated state-representation, we aggregate all those $(s,a)$ in the same block if the differences of their $Q_h^*$ values are less than $\epsilon$ according to definition 1.
>
> 6. Here in (4) randomization is applied to the value function estimates by perturbing observed rewards to drive exploration, which is essential to randomized LSVI algorithm (see [4] for more details).
>
> 7. As we responded in the 2nd point, $\xi_n$ is a parameter carefully derived based upon the concentration bound in Lemma 8 in Appendix B, to make Lemma 9 hold, and to control the terms by a tight upper bound in the proof. This is a technical term and is important to make the algorithm and the proof work.
>
> 8. We thank the reviewer for the suggestion about more future work. We will extend this in a revised version later. For example, we plan to implement our algorithm on real-world data, and derive concurrent RL results on function approximation beyond the current tabular setup, and compare our results with the existing concurrent functional approximation literature (e.g. [5,6]).
>
> [1] Russo, Daniel. "Worst-case regret bounds for exploration via randomized value functions." Advances in Neural Information Processing Systems 32 (2019).
>
> [2] Agrawal, Priyank, Jinglin Chen, and Nan Jiang. ``Improved worst-case regret bounds for randomized least-squares value iteration." Proceedings of the AAAI Conference on Artificial Intelligence, vol. 35, no. 8, 2021.
>
> [3] Bertsekas, Dimitri, and John N. Tsitsiklis. Neuro-dynamic programming. Athena Scientific, 1996.
>
> [4] Osband, Ian, et al. "Deep exploration via randomized value functions." Journal of Machine Learning Research 20.124 (2019): 1-62.
>
> [5] Desai, Nishant, Andrew Critch, and Stuart J. Russell. "Negotiable reinforcement learning for pareto optimal sequential decision-making." Advances in Neural Information Processing Systems 31 (2018).
>
> [6] Min, Yifei, et al. "Cooperative multi-agent reinforcement learning: asynchronous communication and linear function approximation." International Conference on Machine Learning. PMLR, 2023.

---

### Official Review · Reviewer_n92W · 2025-03-13

**Overall Recommendation:** 4

**Summary:**

The paper presents a novel approach to concurrent reinforcement learning (RL) using Randomized Least Squares Value Iteration (RLSVI) with aggregated states. The authors propose a framework where multiple agents interact with a common environment, sharing their experiences to improve decision-making collectively. The paper also discusses the role of the discount factor in RL, arguing that it serves primarily as a tuning parameter to stabilize value updates rather than being an intrinsic part of the environment.

**Claims And Evidence:**

See the comments below

**Essential References Not Discussed:**

See the comments below

**Experimental Designs Or Analyses:**

See the comments below

**Methods And Evaluation Criteria:**

See the comments below

**Other Comments Or Suggestions:**

I suggest adding a ​systematic comparison table to explicitly contrast the proposed approach with existing concurrent RL methods, highlighting theoretical guarantees, empirical performance, and practical considerations to emphasize its novelty and superiority.

**Other Strengths And Weaknesses:**

Strengths:

- The paper addresses an important problem in RL, particularly in the context of multi-agent systems.

- The theoretical results are novel and provide improvements over existing bounds.

- The empirical results are consistent with the theoretical predictions, providing strong evidence for the efficacy of the proposed algorithms.

Weaknesses:

- The empirical validation is limited to synthetic environments. It would be beneficial to see how the algorithm performs in more complex, real-world scenarios.

- The paper could benefit from a more detailed discussion of the practical implications of the theoretical results, particularly in terms of scalability and computational efficiency.

**Questions For Authors:**

- How does the proposed algorithm compare to other concurrent RL algorithms in terms of both theoretical guarantees and empirical performance?

**Relation To Broader Scientific Literature:**

See the comments below

**Theoretical Claims:**

See the comments below

---

> ### Author Rebuttal · Authors · 2025-03-30
>
> We thank the reviewer for the positive feedback and constructive comments.
>
> 1. We would like to emphasize two points regarding the role of experiments in this work. First, the primary contribution of this paper is theoretical analysis of concurrent model-free RLSVI algorithms. Notably, neither of the two most closely related works [1,2], which focus on single-agent RLSVI, includes numerical results. Second, our experiments using synthetic data are explicitly designed to validate theoretical findings in a controlled environment, a common practice in theoretical literature. Importantly, unlike [1,2], our theoretical results are derived from practical, storage-efficient algorithms, and the corresponding numerical validations provide valuable insights for practitioners. Nevertheless, we agree with the reviewer that extending experiments to real-world datasets would be an interesting direction for future exploration, which we plan to pursue.
>
> 2. For scalability, as detailed in lines 248–258 (paragraph under equation (6)), prior approaches [1,2] require storing all historical data, making them computationally impractical at scale, while ours is storage-efficient. Following the reviewer's suggestion, we provide a comparison table below to explicitly contrast our approach with existing related work, including the two most closely related work [1,2] on single-agent RLSVI:
>
> | Agent | Setup        | Algorithm                   | Regret Bound                         | Regret-Type | Data Stored  | Numerical |
> |-------|--------------|-----------------------------|--------------------------------------|-------------|--------------|-----------|
> | Single| Tabular      | RLSVI [1]                   | $\tilde{O}(H^3S^{3/2}\sqrt{AK})$                  | Worst-case  | All-history  | N/A       |
> | Single| Tabular      | RLSVI [2]                   | $\tilde{O}(H^{5/2}S\sqrt{AK})$                 | Worst-case  | All-history  | N/A       |
> | Multi | Tabular      | Concurrent RLSVI [3]        | N/A                                  | Bayes       | All-history  | Synthetic |
> | Multi | Linear Functional Approximation  | Concurrent LSVI [4]         | $\tilde{O}(H^2\sqrt{d^3KN})$                       | Worst-case  | All-history  | N/A       |
> | Multi | Linear Functional Approximation  | Concurrent LSVI [5]         | $\tilde{O}(H\sqrt{dKN})$                     | Worst-case  | All-history  | N/A       |
> | Multi | Tabular      | Concurrent RLSVI (**ours-1**)| $\tilde{O}(H^{5/2}\Gamma\sqrt{KN})$        | Worst-case  | All-history  | N/A       |
> | Multi | Tabular      | Concurrent RLSVI (**ours-2**)| $\tilde{O}(H^{5/2}\Gamma K\sqrt{N})$       | Worst-case  | One episode  | Synthetic |
>
> The following discussions will be added to paper:
>
> -  $\Gamma$ denotes the size of aggregated states. This reduces complexity and accelerates learning by focusing on grouped state-action pairs.
>
> - To the best of our knowledge, all existing concurrent RLSVI work focuses on finite-horizon case, while we also cover the infinite-horizon case. The regret bound for the infinite-horizon case is similar to that of the finite-horizon case, by replacing $KH$ with total time steps $T$.
>
> - The methods marked as "N/A" in the "numerical" column store all agents' trajectories at every step, making them computationally infeasible as $N$ grows large. These approaches (including our first finite-horizon algorithm, second-to-last row in the table) provide only theoretical analyses without numerical results.
>
> - Although [3] stores all data empirically, their RLSVI algorithm assumes a known parametric feature matrix, making it simpler to implement than ours. Also they evaluate only Bayes regret—less stringent than our worst-case regret—and provide empirical results exclusively on synthetic data without theoretical guarantees.
>
> - The last row corresponds to storing only the latest episode data, increasing the regret bound by a factor of $\sqrt{K}$ but reduces space complexity by a factor of $K$, making it computationally feasible.
>
> [1] Russo, Daniel. "Worst-case regret bounds for exploration via randomized value functions." Advances in Neural Information Processing Systems 32 (2019).
>
> [2] Agrawal, Priyank, Jinglin Chen, and Nan Jiang. ``Improved worst-case regret bounds for randomized least-squares value iteration." Proceedings of the AAAI Conference on Artificial Intelligence, vol. 35, no. 8, 2021.
>
> [3] Taiga, Adrien Ali, Aaron Courville, and Marc G. Bellemare. "Introducing Coordination in Concurrent Reinforcement Learning." ICLR 2022 Workshop on Gamification and Multiagent Solutions.
>
> [4] Desai, Nishant, Andrew Critch, and Stuart J. Russell. "Negotiable reinforcement learning for pareto optimal sequential decision-making." Advances in Neural Information Processing Systems 31 (2018).
>
> [5] Min, Yifei, et al. "Cooperative multi-agent reinforcement learning: asynchronous communication and linear function approximation." International Conference on Machine Learning. PMLR, 2023.

---

### Official Review · Reviewer_zmC5 · 2025-03-22

**Overall Recommendation:** 3

**Summary:**

The paper adapts the concurrent learning framework via randomized least squares value iteration with an aggregated state representation, to improve exploration efficiency and the worst-case regret bound. Extensive experiments are conducted to support the theoretical findings.

**Claims And Evidence:**

The claims are well supported by evidence.

**Essential References Not Discussed:**

None.

**Experimental Designs Or Analyses:**

Yes, the experimental designs are valid.

**Methods And Evaluation Criteria:**

The methods and evaluation make sense to me.

**Other Comments Or Suggestions:**

None.

**Other Strengths And Weaknesses:**

Strength
- The paper focuses on an important problem in RL, with sound proof of the worst-case regret bound over the existing ones.

Weakness
- The experiments are only on the synthetic datasets. With experiments on the real-world datasets, it could demonstrate the improvements better.

**Questions For Authors:**

None.

**Relation To Broader Scientific Literature:**

The paper is related to the concurrent randomized least-squares value iteration literature.

**Theoretical Claims:**

Yes. The proofs are correct.

---

> ### Author Rebuttal · Authors · 2025-03-30
>
> We thank the reviewer for the positive feedback and constructive comments. We are aware that our experiments are only on synthetic data. We would like to emphasize two points regarding the role of experiments in this work.
>
> Firstly, the main focus and contribution of our work is the first theoretical analysis of concurrent model-free RLSVI algorithms. It is worth noted that neither of the two most closely related papers [1,2] (single-agent RLSVI) includes any numerical results.
>
> Secondly, we conduct experiments on synthetic data to specifically validate the theoretical findings in a controlled manner. This approach aligns with existing literature of theory papers. It is also worth noted that, unlike [1,2], our theoretical results are based on practical, storage-efficient algorithms. With carefully designed numerical validations, they provide valuable insights for practitioners.
> We will add a table comparing with existing theoretical results to discuss the practical implications in terms of scalability and computational efficiency.
>
> That said, we agree with the reviewer that experiments on real-world data would be an interesting direction to extend the current work, and we plan to explore it in the future.
>
> [1] Russo, Daniel.``Worst-case regret bounds for exploration via randomized value functions." Advances in Neural Information Processing Systems 32 (2019).
>
> [2] Agrawal, Priyank, Jinglin Chen, and Nan Jiang.``Improved worst-case regret bounds for randomized least-squares value iteration." Proceedings of the AAAI Conference on Artificial Intelligence. Vol. 35. No. 8. 2021.

---

### Decision · Program_Chairs · 2025-05-01

**Decision:**

Accept (poster)

**Comment:**

This paper proposes a concurrent reinforcement learning framework based on randomized least-squares value iteration (RLSVI) in multi-agent systems to address the problem of exploration. It demonstrates improvements in efficiency and provides theoretical worst-case regret guarantees, supported by synthetic experiments.

All reviewers agree on the clear motivation and contribution of the proposed method, the soundness and clarity of the theoretical analysis, the validity of the experimental setup, and the overall rigor of the presentation. A common concern was the lack of experiments on real-world tasks and the need for more detailed discussion on the implications and comparisons of the results. The authors addressed these points by explaining the rationale behind using controlled experiments and providing a systematic comparison with prior works. Reviewer 35te also raised several technical questions, which were adequately addressed in the rebuttal.

The authors' response was thorough and resolved most of the reviewers' concerns. Given its novelty, technical depth, and relevance, this paper would be a valuable addition to ICML.